# The Development of a Water Resource Monitoring Ontology as a Research Tool for Sustainable Regional Development

Assel Ospan [1,*], Madina Mansurova [1], Vladimir Barakhnin [2,3], Aliya Nugumanova [4] and Roman Titkov [3]

1 Department of Artificial Intelligence and Big Data, Faculty of Information Technology, Al-Farabi Kazakh National University, Almaty 050040, Kazakhstan; madina.mansurova@kaznu.edu.kz
2 Federal Research Center for Information and Computational Technologies, 630090 Novosibirsk, Russia; bar@ict.nsc.ru
3 Department of Informatics Systems, Faculty of Information Technology, Novosibirsk State University, 630090 Novosibirsk, Russia; r.titkov@g.nsu.ru
4 Department of Big Data and Blockchain Technologies, Astana IT University, Astana 010000, Kazakhstan; a.nugumanova@astanait.edu.kz
* Correspondence: assel.ospan@kaznu.edu.kz; Tel.: +7-708-151-22-39

**Abstract:** The development of knowledge graphs about water resources as a tool for studying the sustainable development of a region is currently an urgent task, because the growing deterioration of the state of water bodies affects the ecology, economy, and health of the population of the region. This study presents a new ontological approach to water resource monitoring in Kazakhstan, providing data integration from heterogeneous sources, semantic analysis, decision support, and querying and searching and presenting new knowledge in the field of water monitoring. The contribution of this work is the integration of table extraction and understanding, semantic web rule language, semantic sensor network, time ontology methods, and the inclusion of a module of socioeconomic indicators that reveal the impact of water quality on the quality of life of the population. Using machine learning methods, the study derived six ontological rules to establish new knowledge about water resource monitoring. The results of the queries demonstrate the effectiveness of the proposed method, demonstrating its potential to improve water monitoring practices, promote sustainable resource management, and support decision-making processes in Kazakhstan, and can also be integrated into the ontology of water resources at the scale of Central Asia.

**Keywords:** knowledge graph; ontology; semantic web; water resource monitoring; spatial data; RDF triples

## 1. Introduction

Water resources play an important role in human life and ecosystems, and their effective monitoring and management are essential for sustainable development. Thus, the resolution adopted by the UN General Assembly in 2016, "International Decade for Action "Water for Sustainable Development", 2018–2028" [1], emphasizes the need for increased attention to sustainable development and integrated water resource management in order to achieve socioeconomic and environmental objectives and the implementation and promotion of relevant programs and projects.

The monitoring of the Ili-Balkhash basin (hereinafter IBB) is extremely important as the largest lake ecosystem and a source of valuable natural resources, because almost 40% of all hydropower resources of Kazakhstan are concentrated in the IBB. The basin of Lake Balkhash occupies a vast territory and covers several regions of Kazakhstan and China [2]. In the Kazakh part of the basin, there is the city of Almaty—a large metropolis with a population of 2.5 million. According to environmentalists, Lake Balkhash is threatened by an ecological catastrophe: its main tributary, the Ili River, is becoming smaller every year due to a decrease in water volume [3].

In light of the above, there is an obvious need to create an intelligent system for monitoring the water resources of the Ili-Balkhash basin in order to gain new knowledge about the state of water use facilities and make informed management decisions based on them. When creating such systems, it becomes necessary to develop technologies capable of processing and analyzing large volumes of semantically related data on water resources. One of the key features of this work is the use of a variety of data sources, including physical sources, sensors, web sources, official reports, documents, and other information resources in the field of water consumption and ecology [4–8]. The use of heterogeneous data sources makes it possible to obtain more complete and reliable information about the quality, quantity, and condition of water resources.

Because water monitoring consists of many parameters, such as level, flow, quality, and chemical composition, it was necessary to organize the processing of these parameters in such a way as to ensure the integration of water data for effective searches, semantic analysis, and reasoning. Thus, it was decided to develop a knowledge graph for this subject area [9]. A knowledge graph is a universal information representation structure that models data as interconnected nodes and edges, forming a graph-like structure. Knowledge graphs facilitate the organization, storage, and retrieval of structured data in a way that reflects the internal relationships and semantics of information. This framework provides advanced data querying, reasoning, and inference, making knowledge graphs an invaluable tool for various applications, such as information retrieval, recommender systems, semantic search, and data integration [10].

The knowledge graph for monitoring water resources has the following advantages:

- It provides a framework for integrating data from various sources, including sensors, monitoring stations, databases, and external water body datasets;
- It allows one to capture contextual information, which presents spatial and temporal measurements of water level and flow, water quality parameters, and hydrological factors, which provide an opportunity to obtain a comprehensive understanding of the complex interactions and dynamics in the aquatic ecosystem;
- It displays the semantic relationships between water resource objects, which allows for deep data analysis, facilitating queries and predicting the consequences of changes in the state of waters;
- It detects factors influencing water quality or level, allowing one to explore each node and the connections between them to identify hidden patterns and anomalies;
- It provides a decision support system related to the monitoring of water resources, making it possible to generate recommendations with the help of subject-matter experts;
- It is easily scalable and flexible, which allows it to be expanded and adapted as new data become available or monitoring requirements or parameters change.

The knowledge graph is built using ontology and semantic models. This paper presents an ontology model based on the SWRL [11] language, which provides the ability to express complex relationships and derive new knowledge in ontologies. The use of the technology of the semantic sensor network (SSN) [12] makes it possible to interact with data from sensors and observations, ensuring the integration and processing of these data. In addition, table extraction and understanding methods are used to obtain valuable information from PDF flies. Time ontology [13] is used, which models and analyzes information related to temporal indicators, which is especially important in the context of water resource monitoring.

Thus, the purpose of this work is the semiautomatic creation of an ontology of the water resources of the Ile-Balkhash water basin based on heterogeneous data obtained from both web sources and sensory sources. The contribution of this work to the problems related to the semiautomatic construction of ontologies of water resources consists of three points. Firstly, we present a platform for the automatic extraction of geodata from web tables, the sources of which are the official websites of the Republic of Kazakhstan, with norms and laws regarding water use, Wikipedia, and the sites of ecological water posts. The platform extracts data from heterogeneous sources using two tools: (1) Table Processor,

extracting data from Excel tables and (2) Qurma, extracting data from web tables and PDF files. The received data are semantically aligned using a special platform module and distributed over five loosely connected knowledge graphs that form a single ontology of water resources. The constructed and published ontology of the Ile-Balkhash water basin is our second contribution to the problem. Finally, the third contribution is the development of a software decision support module, which currently relies on six basic rules, allows users to form semantic queries regarding the ontology and obtain answers to current questions, informs users about the current state of water resources, and proactively responds to potential environmental threats.

The further structure of the work is as follows: Section 2 contains a list of related works. Section 3 briefly characterizes the study area. Section 4 contains a description of data sources and methods for their processing, including ontology construction methods, as well as their structure (modules, properties, and rules). Section 5 formulates the main results of the work. Section 6 contains a discuseession to conduct a general analysis of the purpose of this work, and Section 7 contains the conclusion, which presents plans for further research.

## 2. Related Works

The application of an ontology for monitoring water resources is successfully practiced all over the world.

Ontology, in the context of information science and knowledge representation, refers to a formal and structured framework for modeling and representing knowledge about a specific domain. It involves defining a set of concepts, entities, attributes, and relationships that collectively describe the essential elements and their interconnections within the domain. The primary goal of an ontology is to establish a shared and standardized understanding of the domain, enabling effective communication, data integration, reasoning, and analysis across various systems and applications. A well-defined ontology provides a systematic way to represent not only the entities themselves but also their properties, relationships, and constraints. This structured representation enables computers to interpret and process information with greater accuracy and efficiency, ultimately enhancing the ability to extract insights, make informed decisions, and perform automated reasoning tasks within the domain [10].

The authors of [14] proposed a semantic web method for modeling ontology and constructing rules for monitoring river water quality, which also used observations from sensors. As a result of their work, a system was obtained to track the entire process of pollution events and warn of approaching epidemiological values. This work was very helpful in developing the third module of our proposed water monitoring ontology (see Section 4.3).

Before creating an ontology, it was necessary to reformat data from heterogeneous sources into a machine-readable format; in our case, to replenish the knowledge graphs, the triplet format (XML, RDF, and JSON) was required. This practice was successfully applied in [15], where the authors presented an architecture for integrating heterogeneous sources into a unified knowledge graph, from which useful knowledge about water quality was successfully extracted by analyzing the physicochemical and biological properties using spatiotemporal values and normative documents about water quality. The methods outlined in this work allowed us to develop our own architecture for the process of creating a water monitoring ontology, which we expanded with three methods of data parsing and the inclusion of additional data on the level and flow of water (see Section 4.3).

Sensor data play a key role in monitoring water objects, so the SSN ontology provides interaction with sensor and observation data. A successful application of the SSN ontology is presented in [16], where an ontology was developed using data from hydrological sensors, by importing time ontology and instantiating hydrological classes and establishing reasoning rules. Based on this work, we developed Module 2.

The SWRL and SSN methods acted as the basis for creating linked data, because these methods allow the user to create complex relationships and derive new knowledge from the ontology. The SWRL method has been successfully applied in solving problems from other areas, such as diagnosing and treating vector-borne diseases [17], solving the problem of knowledge sharing about complex engineering systems for diagnostics [18], and facilitating preventive maintenance in industry using fuzzy clustering. Also, in [19] and [20], the results of the successful use of SSN are presented.

For several years, we have been monitoring the dynamics of the width of the Ili River on the border with China using remote sensing images from Landsat and Sentinel, the results of which were published in [21]. Long-term analysis shows that since 1980, the width of the Ili River has decreased from 400 m to 270 m, which once again proves the decrease in the water level of the river. The results of this work prompted us to continue the analysis and create a unified system for monitoring Ili water resources. Module 4 includes socioeconomic indicators in the regions fed by the Ili River, including disease statistics, because in [22], the authors present their evidence of the impact of surface water quality on diseases of the population living near water objects. A correlation was also found between the growth of the population of the Almaty region and the decrease in the quantity/deterioration of the quality of waters in the IBB.

Because one of the main contributions of our work is the generation of ontology based on web tables, it is important to point out previous work focused on extracting ontology from tabular sources. Most of the early work in this area is based on heuristics that govern how elements of web tables are recognized and classified and mapped to classes and other ontology elements.

One of the early works in this area is [23], which presents a tool for automatically generating ontologies from regular web tables. As the authors note, the process resembles the re-engineering of relational tables into conceptual models but takes into account a wider range of table layout patterns based on heuristics. This work is useful in that in addition to presenting heuristic rules, it presents a report on typical recognition errors, such as the duplication of concepts, the incorrect labels of concepts, and the recognition of aggregating concepts.

In [24], a heuristic approach is also proposed, which is used to automatically extract the ontology of the selected domain based on the headers of web tables. The construction of the ontology takes place in two stages: (a) extracting the table schema as a pseudo-ontology from each table of the domain and (b) building a domain ontology that combines these extracted table schemas. Schemas are merged by splitting and clustering using statistical and heuristic information that takes into account the structural and semantic characteristics of web table headers. Ref. [25] presents an automatic method for generating RDF from web tables of various shapes and types. This paper defines six representative types of web tables commonly used in web applications, and for each type, it uses a separate heuristic approach to convert tables of this type to RDF.

Ref. [26] considers a large class of web tables containing geographic data that are suitable for integrating big data into various applications, such as urban resilience, transportation networks, political research, and public health. The semantic extraction of geodata has its own characteristics related to the need to correctly display the conceptual hierarchy and determine the metadata associated with instances, the geographic information corresponding to the properties of ontology elements, and the cell values that can be used to determine geographic coordinates. The authors of the work propose an architecture, the central modules of which are geocoding and disambiguation of toponyms.

Ref. [27] highlights the following limitations of modern approaches to the semantic annotation of web tables: (1) the lack of use of contextual semantics; (2) the reliance on external metadata, for example, on descriptions of tables, which are not always available for real tables; (3) the introduction of specific knowledge graphs, which prevents the generalization of approaches to the annotation; (4) the identification of only a subset of possible annotations; and (5) the lack of implementations and working tools. To overcome these

limitations, the authors propose a comprehensive approach and a tool called MantisTable, which provides an unsupervised and fully automated approach to table annotation even for tables without headers or other external information using knowledge graphs. The approach proposed by the authors consists of eight steps: (1) data processing and normalization, (2) column analysis and subject definition, (3) data extraction, (4) cell entity annotation, (5) column predicate annotation, (6) column type annotation, (7) revision, and (8) exporting.

In this work, we also build an information system for extracting data from web tables, taking into account and overcoming these limitations (see Table 1). As follows from the table, the difference between our work and [28] is that we used contextual information not only from the table itself, but also by reading textual information before or after the table. We identified the resulting contexts using thematic modeling methods and word embeddings.

**Table 1.** Comparison of methods for extracting data from web tables.

| Limitation | Way to Overcome the Limitation, Represented in [27] | Proposed Way to Overcome the Limitation (Qurma + TableProcessor) | Example |
|---|---|---|---|
| The lack of contextual information | The Semantic table interpretation (STI) engine calculates a measure of similarity for all embedded concepts and contexts. | The Qurma tool [29] uses data from Wikipedia, which looks at which topic the term is most likely to refer to. | Qurma enables automatic Wikipedia searches to uncover the shared concept: 'Rivers flowing into Balkhash'. |
| Dependency on table metadata | STI uses a similarity metric, such as cosine similarity or dot product. | The Qurma tool uses 10 heuristics to avoid relying on metadata | An example of one of the heuristics is the rule for automatic recognition of coordinates. |
| Dependence on specific knowledge graphs | TableMiner+ utilizes OpenIE for domain-independent structured information retrieval. | The Qurma tool extracts tables across domains using the CleanArchitecture method. | The tool searches the entire Wikipedia and can retrieve data from any subject area. |
| Identification of a limited set of annotations | STILTool assesses annotations from various approaches on T2Dv2 and Limaye200 Gold Standard tables. | The TableProcessor [25–30] tool, having received a new concept, adds it to the system, and models the semantic interdependence. | New names, previously unknown to the system, are placed in a special buffer zone and offered to the user for manual confirmation or rejection. |
| Lack of implementation | MantisTable implements two modern approaches (T2Dv2 and Limaye200) and compares with a baseline method that uses different annotation steps for concepts, data types, and predicates. | The QURMA system's Fluent Design interface allows users to specify a document's URL for table retrieval. TableProcessor has a ready-made script on GitHub where any user can process their table. | 1. Qurma: https://github.com/Kyrmasch/Sorge (accessed on 6 July 2022) 2. Table Processor: https://github.com/Igriva/TableProcessor (accessed on 27 May 2022) |

Finally, a large number of works have appeared recently in connection with the strengthening of the role of machine learning in solving various problems of semantic modeling [31–37]. These works demonstrate high-level results but require a large amount of training data.

## 3. Study Area

The Ili-Balkhash basin is one of the largest lake ecosystems on the planet and is a unique natural complex that occupies a vast territory of 413 thousand km$^2$, 85% of which (353 thousand km$^2$) is located on the territory of Kazakhstan and 15% on the territory of China. The Kazakhstan part of the Ili-Balkhash basin includes the territory of the Almaty, Zhambyl, and Karaganda regions. There is a large metropolis in the basin, the city of Almaty, with a population of 2.5 million. Five large rivers flow into Lake Balkhash: Ili (1439 km), Karatal (390 km), Ayaguz (492 km), Lepsy (418 km), and Aksu (316 km). The total volume of potential hydropower resources in the basins of Lake Balkhash is 63.5 billion kWh; almost 40% of all hydropower resources of Kazakhstan are concentrated here, and the potential of the Ili River is 7008 million kWh, which is 18.2% of the potential of all IBB rivers, namely 35.5 billion kWh. One of the active water users in the Ili-Balkhash basin is

irrigated agriculture, which has been practiced here for more than a century; for example, in 1915, the area of irrigated land was 290 thousand hectares [38].

According to environmentalists, Lake Balkhash is threatened by an ecological catastrophe: its main tributary, the Ili River, is becoming smaller every year due to a decrease in the volume of water. Now, we are working with China only on an agreement on water distribution. During the implementation of the two-year EU project on Balkhash, it was calculated that a minimum flow from China of 12 km$^3$ per year should be maintained. But now it has significantly decreased to 8 km$^3$ [39].

At present, the main consumer of water in this basin, both on the territory of the Republic of Kazakhstan and on the territory of China, is irrigated agriculture, for which more than 70% of the water resources are spent [39]. As recent data show, the use of water for irrigated agriculture tends to increase, especially in China, which raises concerns for the safety of Lake Balkhash and for the ecological situation in the basin. The situation is also aggravated by global and regional climate change and the degradation of glaciers in the basin [40].

Therefore, studying the state of the basin in terms of the impact of economic activity on the environment, identifying trends, and developing recommendations for mitigating anthropogenic impact and climate change play an important role in developing a science-based strategy for managing the use and protection of water resources and applying the principles and approaches of Integrated Water Management (IWM).

The problems of IWM implementation in Kazakhstan, as well as the data obtained during this study, are relevant and in demand [39]. This study used data from 63 monitoring stations (Figure 1), where various sensors were installed, and observations that collected data from 1 January 2001 to the present. The observation parameters obtained by these sensors include water level, flow and temperature, information about floods, and state of surface water quality [4,5].

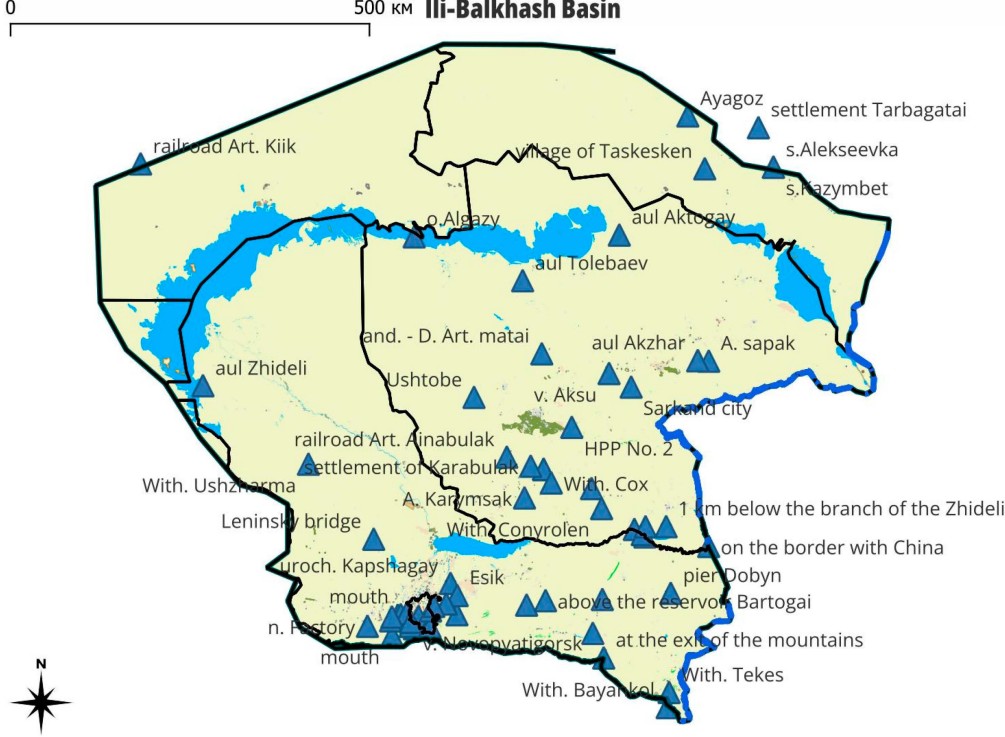

**Figure 1.** The layout of the hydrometric stations of the Ili-Balkhash basin.

## 4. Materials and Methods

### 4.1. Data Sources and Semantic Alignment

To build an ontology for monitoring the water resources of the IBB, it was necessary to collect data. The problem was that there is no unified database for the water resources of Kazakhstan, so the data were collected from heterogeneous sources and divided into 4 groups: 1. Water Regulations, 2. Sensor Data, 3. Water Objects, and 4. Socioeconomic Indicators.

For the Water Regulations group, we collected water regulation data, which set thresholds for water use and quality, which was measured using the Water Pollution Index (WPI) parameter. According to the WPI parameter, the classes of surface water pollution and their characteristics were determined, as presented in Table A1 (see Appendix A). The data for the table were extracted from the Information and Legal System of Normative Legal Acts of the Republic of Kazakhstan from the paragraph on the Water Code of the Republic of Kazakhstan [7]. The data on this web page are presented in text and tables, as shown in Figure 2, so the water regulation data were extracted using the QURMA tool into a structured format [29].

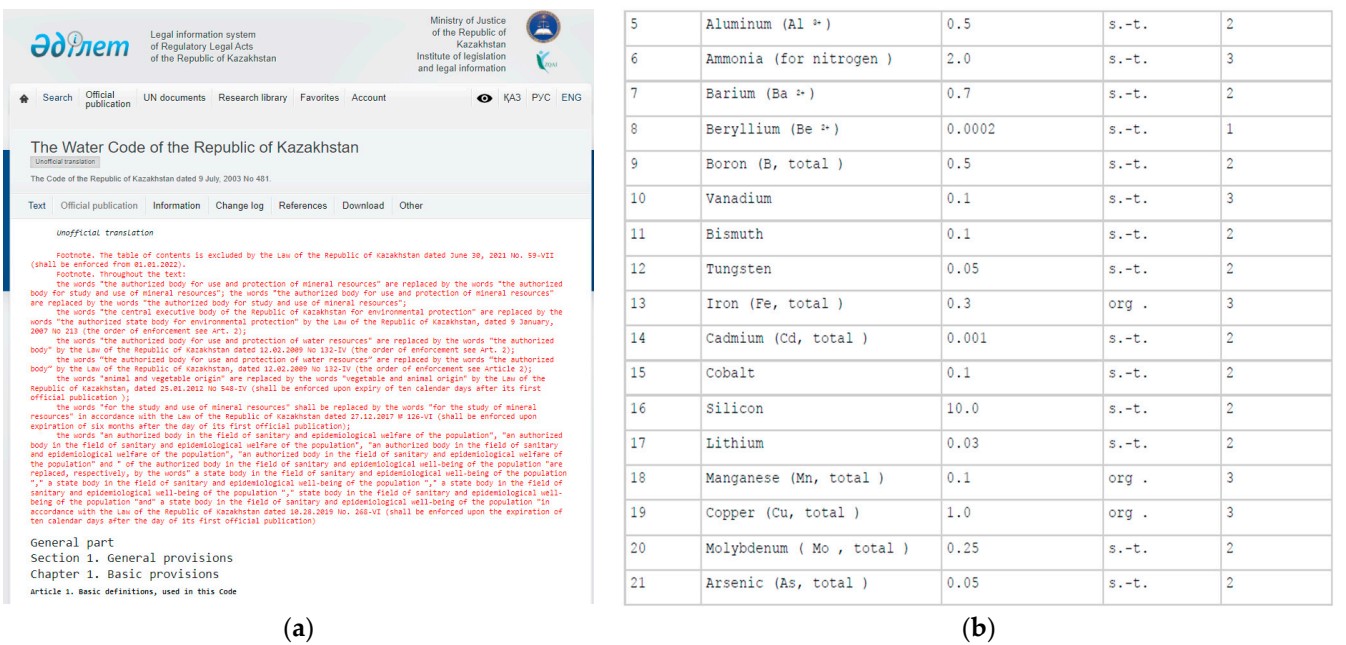

(**a**)                                                                              (**b**)

**Figure 2.** Information and legal system of normative legal acts on the Water Code of the Republic of Kazakhstan: (**a**) web page; (**b**) type of data storage on this web page.

The Sensor Data group was the basis of the knowledge graph of water resource monitoring, as it contained daily/monthly/quarterly observations from sensors about the level, flow, and quality of the water. Records were collected from the installed gauging stations in the basin, and all records from the sensors were uploaded by the National Hydrometeorological Service of the Republic of Kazakhstan [5] to their website in the form of a report in PDF files—examples of these files are presented in Figure 3. A total of 63 hydrostations have been installed in the IBB, which have been collecting data since 1995, dividing data into 2 categories: 1. observations on the level and flow of the water [4] and 2. the state of the quality of the surface waters [5]. Category 1 contains data on the following parameters: location of hydrological posts, water level, state of a water object, water temperature, meteorological conditions, water discharges, ice thickness and snow depth on ice, ice phenomena at the site of the post, and information about floods and rain floods. Here, all data are already recorded in a structured file format (CSV, Excel, and SQL) and stored on the official Ecodata website [4]. Category 2 stores observations of surface water quality in the Almaty region, carried out at 42 gates of 22 water objects

(rivers Ili, Tekes, Korgas, Kishi Almaty, Esentai, Ulken Almaty, Chilik, Charyn, Bayankol, Kaskelen, Karkara, Esik, Turgen, Talgar, Temirlik, Karatal, Aksu, and Lepsi; lakes Ulken Almaty, Alakol, and Balkhash; and reservoir Kapchagay), where 44 physical and chemical indicators of quality are determined: temperature, suspended solids, transparency, pH, dissolved oxygen, BOD5, COD, main ions of salt composition, biogenic elements, organic substances (petroleum products and phenols), heavy metals, and pesticides.

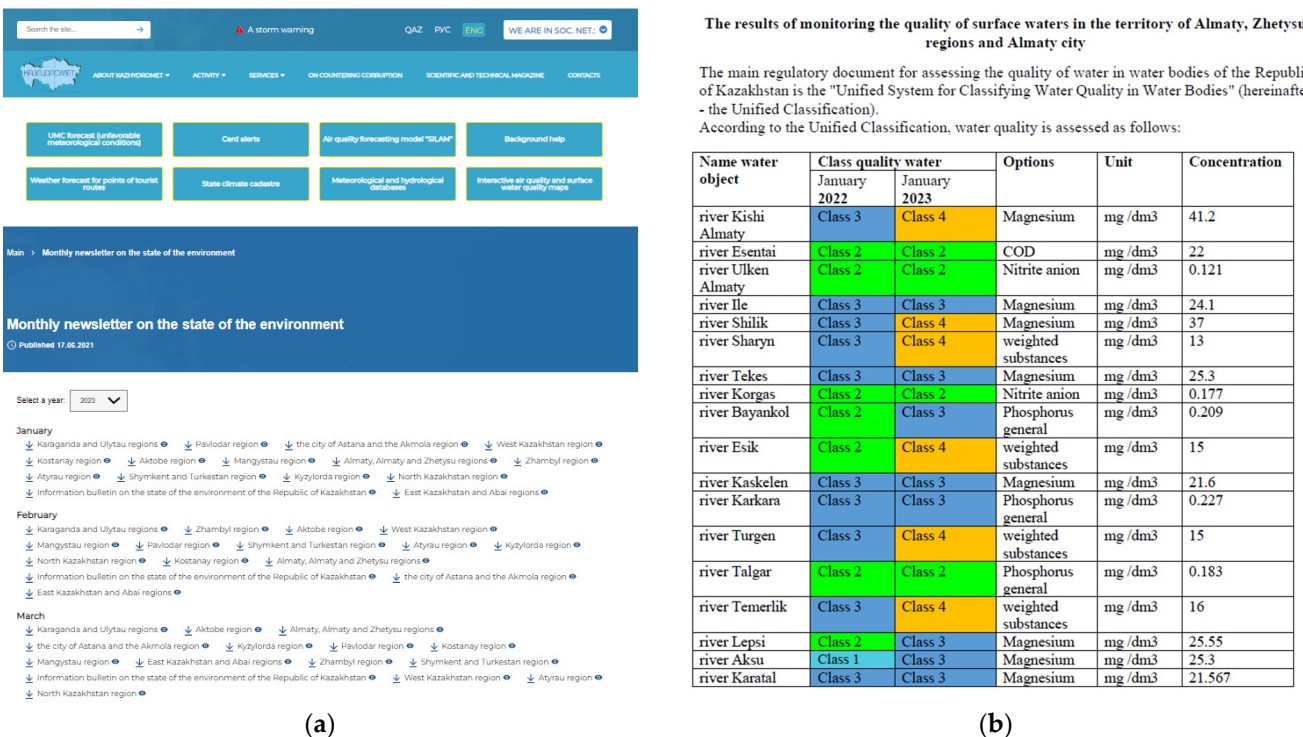

(**a**)                                                                      (**b**)

**Figure 3.** Web page of the National Hydrometeorological Service of the Republic of Kazakhstan: (**a**) monthly newsletter on the state of the environment; (**b**) surface water quality monitoring report for Almaty region for 2022–2023. Here, Class 1 is shown in light blue, Class 2 in green, Class 3 in dark blue, and Class 4 in orange.

The source for the Water Objects group was the open encyclopedia Wikipedia [8], from which general data on the IBB, its rivers and lakes were taken, with data such as physical and geographical description, soils and vegetation, hydrography, glaciers, hydropower resources, and knowledge of river flow and economic activity.

The last group was the Socioeconomic Indicators group which contained information about the regions where the water bodies of the basin are located, obtained from the Bureau of National Statistics of the Republic of Kazakhstan [6]. Here, all statistical records were stored in Excel and text files, as shown in Figure 4. Because there are many socioeconomic indicators, indicators were chosen that are influenced by the resources and quality of nearby water objects. Thus, the following indicators were chosen: population, birth rate/mortality, diseases of the circulatory system associated with iodine deficiency, malignant neoplasms, acute infections of the upper respiratory tract, life expectancy, and types and activities of industrial companies. Table A2 presents 4 groups of sources and the contents of the indicators (see Appendix A).

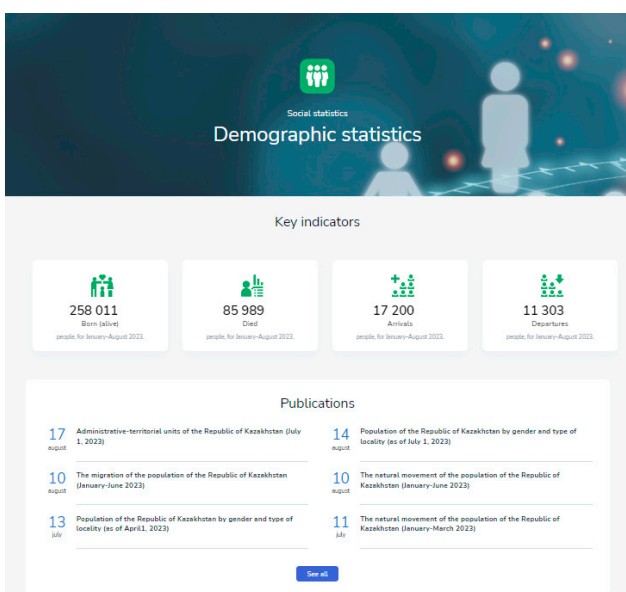

| | 1. Population of the Republic of Kazakhstan as of July 1, 2023 | | | | | | |
|---|---|---|---|---|---|---|---|
| 1 | | | | | | | |
| 2 | | | | | | | |
| 3 | | Total population | | | | Including | |
| 4 | | | Men | Women | Urban population | Including | |
| 5 | | | | | | Men | Women |
| 6 | **Republic of Kazakhstan** | 19,899,377 | 9,715,592 | 10,183,785 | 12,317,671 | 5,867,855 | 6,449,816 |
| 7 | Abai | 609,967 | 298,887 | 311,08 | 371,968 | 178,646 | 193,322 |
| 8 | Akmola | 788,677 | 385,801 | 402,876 | 443,831 | 211,434 | 232,397 |
| 9 | Aktobe | 934,061 | 458,234 | 475,827 | 697,82 | 337,05 | 360,77 |
| 10 | Almaty | 1,518,282 | 759,073 | 759,209 | 245,431 | 119,674 | 125,757 |
| 11 | Atyrau | 698,781 | 345,15 | 353,631 | 386,053 | 186,878 | 199,175 |
| 12 | Batys Kazakhstan | 691,123 | 337,943 | 353,18 | 389,262 | 184,992 | 204,27 |
| 13 | Zhambyl | 1,222,257 | 606,481 | 615,776 | 529,069 | 254,797 | 274,272 |
| 14 | Zhetisu | 699,192 | 345,525 | 353,667 | 310,957 | 148,896 | 162,061 |
| 15 | Karagandy | 1,136,039 | 543,895 | 592,144 | 924,496 | 435,998 | 488,498 |
| 16 | Kostanai | 831,593 | 402,166 | 429,427 | 516,09 | 242,259 | 273,831 |
| 17 | Kyzylorda | 838,86 | 421,246 | 417,614 | 393,694 | 193,054 | 200,64 |
| 18 | Mangystau | 777,205 | 386,503 | 390,702 | 352,253 | 170,729 | 181,524 |
| 19 | Pavlodar | 755,424 | 362,875 | 392,549 | 533,6 | 249,905 | 283,695 |
| 20 | Soltustik Kazakhstan | 532,766 | 257,394 | 275,372 | 259,343 | 120,528 | 138,815 |
| 21 | Turkistan | 2,133,559 | 1,086,853 | 1,046,706 | 523,793 | 262,568 | 261,225 |
| 22 | Ulytau | 221,747 | 108,572 | 113,175 | 175,549 | 84,661 | 90,888 |
| 23 | Shygys Kazakhstan | 729,35 | 349,185 | 380,165 | 483,968 | 225,977 | 257,991 |
| 24 | Astana city | 1,383,291 | 659,083 | 724,208 | 1,383,291 | 659,083 | 724,208 |
| 25 | Almaty city | 2,191,314 | 1,018,163 | 1,173,151 | 2,191,314 | 1,018,163 | 1,173,151 |
| 26 | Shymkent city | 1,205,889 | 582,563 | 623,326 | 1,205,889 | 582,563 | 623,326 |

(**a**)　　　　　　　　　　　　　　　　　　　　　　　　　　　　(**b**)

**Figure 4.** Web page of the Bureau of National Statistics of the Republic of Kazakhstan; (**a**) web page with uploaded reports; (**b**) Excel files of statistics data for 2023.

After collecting data from heterogeneous sources, it was necessary to organize them so that they are semantically related to each other to create an ontology. Thus, we developed an architecture for organizing data into knowledge graphs. The architecture is shown in Figure 5, where the left column shows our heterogeneous data sources, which are divided into 4 groups:

1. Water Regulations—consists of the regulatory rules of the Water Code of the Republic of Kazakhstan. All data are presented on web pages (.html format).
2. Sensor Data—data from sensors that are installed at the gauging stations of the basin and contain information from 1995 on days, months, and years. All sensor records are presented in the form of reports (PDF format) on the website of the National Hydrometeorological Service of the Republic of Kazakhstan [5].
3. Water Objects—information about water objects of the IBB, which includes general data, such as the volume of water and the length of rivers. All data are taken from the open encyclopedia Wikipedia (.html format).
4. Socioeconomic Indicators—indicators from the Bureau of National Statistics of the Republic of Kazakhstan, namely those that are affected by the quality and quantity of water.

Next, we observed arrows that pointed toward triplets in XML. Here, we needed to pay attention to the records of arrows—these are the methods by which data were derived from non/semistructured sources in the form of triplets: object–attribute–value. The first method was TableProcessor [30], developed by us, which uses semantic analysis to extract useful information from web pages and save the data in a machine-readable form, in JSON format. Our second method was Qurma [29]. This method extracts useful knowledge from PDF files to replenish the knowledge base, and all data are extracted in triplet format. The third tool, TableMiner [41], is designed to extract data from Excel files, which performs semantic analysis and is based on machine learning.

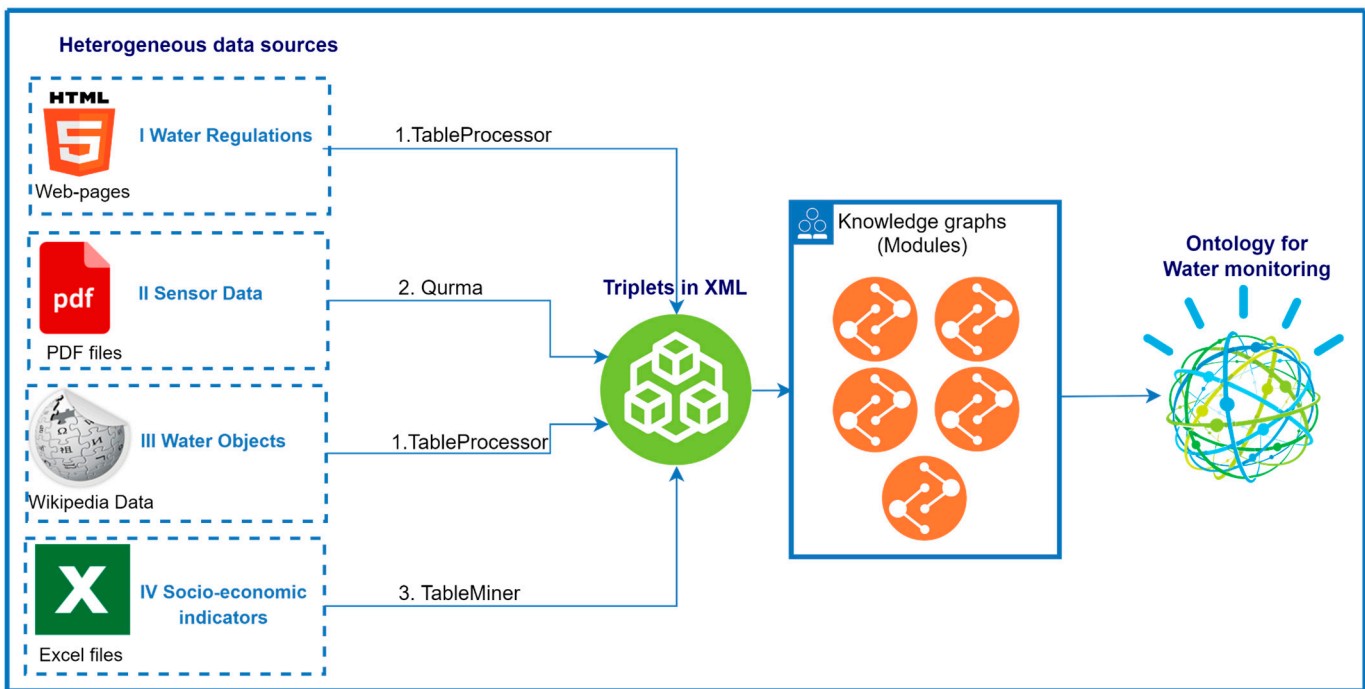

**Figure 5.** Architecture of the process of creating an ontology of monitoring water resources.

All three of these methods were used to represent all of our data as triplets in XML format, as shown in Figure 6, in order to further build an ontology. After presenting all of the data in the form of triplets, a knowledge graph was built based on logical connections and relationships. All logical connections and ways of obtaining new knowledge are described in the next subsection, Ontology Modules.

```xml
<?xml version="1.0" encoding="UTF-8"?>
<water_quality>
  - <row>
        <index>0</index>
        <Rivers_KZ>Irtysh </Rivers_KZ>
        <Type_of_water_objects>river</Type_of_water_objects>
        <Regions>East Kazakhstan</Regions>
        <WPI__April_2005>3.13</WPI__April_2005>
        <WPI__March_2006>2.3</WPI__March_2006>
        <WPI__April_2006>2.35</WPI__April_2006>
        <Ingredients_and_indicators_of_water_quality>Copper</Ingredients_and_indicators_of_water_quality>
        <Average_concentration>0.0019</Average_concentration>
        <Multiplicity_of_exceeding_the_MPC>1.9</Multiplicity_of_exceeding_the_MPC>
        <Classes>III class </Classes>
        <Water_quality_characteristic>"moderately polluted"</Water_quality_characteristic>
  </row>
```

**Figure 6.** Representation of water quality data as triplets in XML format.

### 4.2. Methods for Creating an Ontology and Its Population

The proposed ontology of water resource monitoring was created by applying the SWRL and SSN semantic network methods. The SWRL method based on the integration of OWL and RuleML was proposed in 2019, and now this method is used to solve many problems related to semantic web reasoning, ontology-based knowledge systems, rule-based inference, and structured knowledge integration from various sources [11].

SWRL rules consist of an antecedent (also called a body) and a consequent. The antecedent indicates the conditions that must be met for the rule to be applicable, and the

consequent indicates new information or conclusions that can be drawn when the rule is triggered; i.e., if the antecedent is true, then the consequent must also be true:

rule:: = ′Implies(′ [ URI reference ] { annotation } antecedent consequent ′)′
antecedent:: = ′Antecedent(′ { atom } ′)′
consequent ::= ′Consequent(′ { atom } ′)′

The semantic sensor network SSN was proposed in 2017 [12], and the main goal of this ontology is the interaction with sensor and observation data. Nowadays, SSN provides integration, detection, and interaction of sensors from various sources.

Also, an important part of the ontology is occupied by time ontology [13], which is used to model and analyze time-related information. Time ontology is very useful for creating our ontology, as it has the following properties: it provides a structured and formal representation of temporal concepts, such as intervals, durations, points in time, and temporal relationships; supports the integration and compatibility of time-related data from different sources; allows the user to model and track events over time; facilitates the analysis and understanding of historical data; supports the formulation and execution of time-based queries on temporal data; and provides a link to the semantic web and related data.

In the process of creating the IBB Water Resource Monitoring Ontology, data were collected from various sources and a thorough analysis was performed to select needed data that would provide an effective ontology. Based on the successful work on creating an ontology for water resources [14–16] and based on SWRL, SSN, and time ontology, indicators were identified that can be used to perform semantic analysis.

Building an ontology consists of seven steps:

1. Determining the subject area and scope of the ontology;
2. Considering the possibility of reusing existing ontologies;
3. Listing important terms in the ontology (main classes);
4. Defining classes and the class hierarchy;
5. Defining class properties;
6. Defining threshold data values;
7. Creating entity instances.

Based on the collected data, an ontology was developed, as shown in Figure 7, which links all the above data, forming a unified system that allows for the monitoring of water resources.

The ontology consists of 5 modules, and these modules are interconnected, allowing us to extract useful data using the necessary objects or time intervals.

I. Water Regulations Module (WR): the development of this module was built using the regulatory rules of the Water Code [7]. This module contains physicochemical and microbiological parameters, sampling points, hydrological units, and associated methods. There are 2 classes defined in this module:

1.1. The *WR:water* class is used to describe the quality of water objects based on water regulations. This class has 5 subclasses, each of which describes pollution classes according to the water pollution index (WPI) (Table A1, see Appendix A).

1.2. The *WR:water hygiene standards* class describes the chemical and microbiological composition of water, on the basis of which water pollution classes are determined. This class consists of 4 data properties: oil products, surfactants (SASs), organic substances, and inorganic substances (cations).

II. Observation Data Module (OD): this module is based on data from sensors that are installed at the hydrostations in the basin and contain information since 1995 by day, month, and year. This module contains 1 superclass Sensor Observation and 2 of its subclasses: surface water resource and water quality:

2.1. The *OD:surface water resource* class contains annual data on the regime and resources of land surface waters and describes water parameters in the time interval, such as codes



of hydrological posts, water level, water temperature, water discharges, ice thickness and snow depth on ice, and information about flood and rain flood.

2.2. The *OD:water quality* class contains monitoring data on surface water quality in the territory of the IBB at 42 gates of 22 water bodies (rivers Ili, Tekes, Korgas, Kishi Almaty, Esentai, Ulken Almaty, Chilik, Charyn, Bayankol, Kaskelen, Karkara, Esik, Turgen, Talgar, Temirlik, Karatal, Aksu, and Lepsi; lakes Ulken Almaty, Alakol, and Balkhash; and reservoir Kapchagai). When studying surface waters, 44 physical and chemical indicators of quality are determined in the taken water samples: temperature, suspended solids, transparency, hydrogen index (pH), dissolved oxygen, BOD5, COD, main ions of salt composition, biogenic elements, organic substances (petroleum products and phenols), heavy metals, and pesticides.

III. Observed Objects Module (OO): this module contains the main water objects in the water economic basin. There is 1 superclass Water objects, which has 3 subclasses: water basins, lakes and reservoirs, and rivers and canals:

3.1. The *OO:water basins* class describes basins that contain subclasses lakes and reservoirs, rivers and canals, and data properties, such as basin area, water resource, and energy resource.

3.2. The *OO: lakes and reservoirs* class also contains area, water, and energy data properties.

3.3. The *OO:rivers and canal* class contains the properties of the data and the length of the rivers, the length of the rivers in Kazakhstan, and where the river flows into.

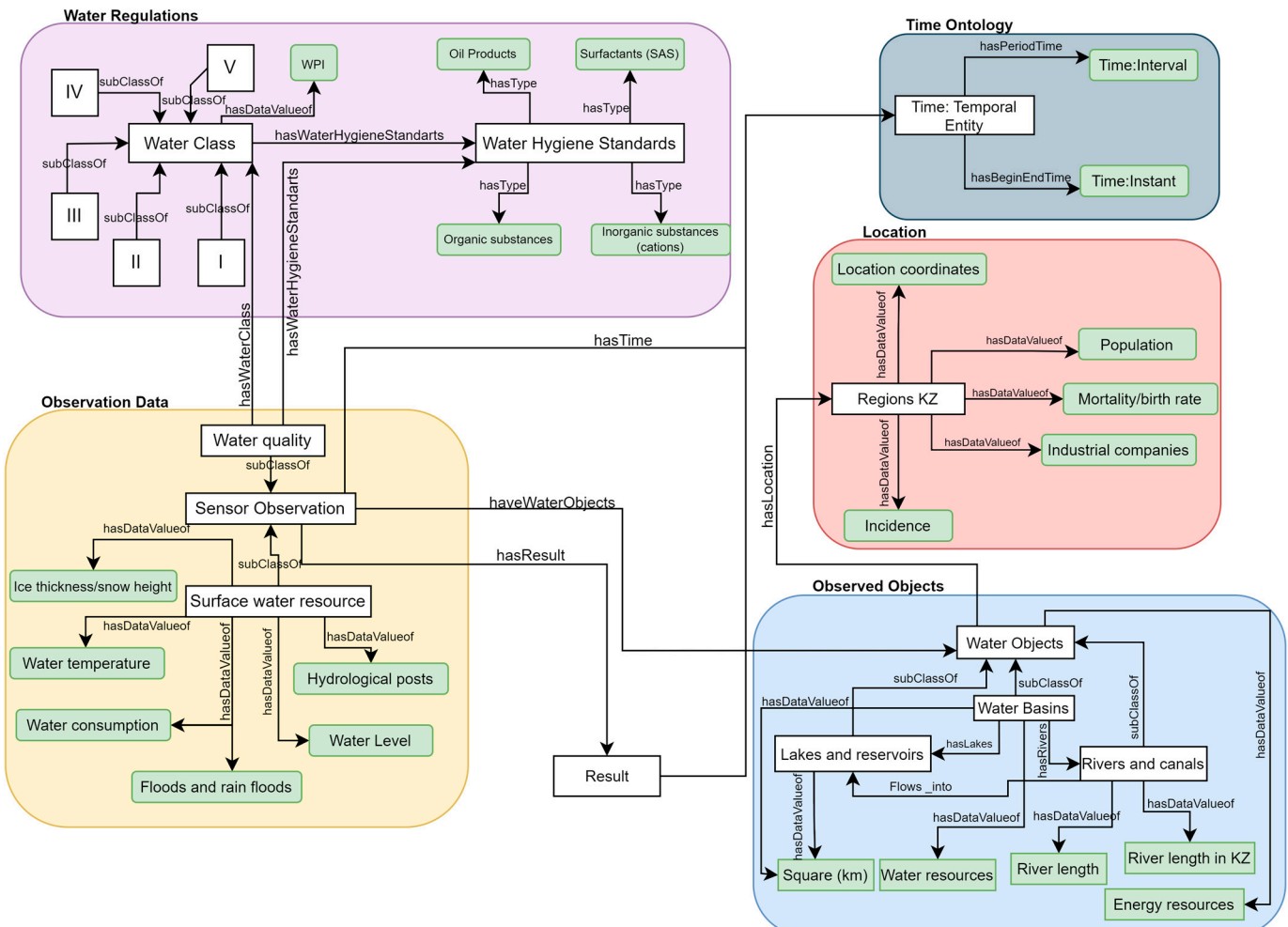

**Figure 7.** Scheme of ontology of water resource monitoring.

IV. Time Ontology Module: Based on the OWL-2 DL ontology of time concepts for describing time properties, which has two subclasses: Time:Interval and Time:Instant.

4.1. *Time:Interval* describes the length in a certain interval.
4.2. *Time:Instant* describes one set time, when start and end must match.

V. Location Module (L): Location Module (L): describes the location, specifically the regions of Kazakhstan because each water body belongs to a region or city. This module consists of 1 main class Regions and its properties:

5.1. *L: Regions* class contains data on all regions and cities of Kazakhstan, and all these regions have coordinates. The Regions class has indicators such as population, births/deaths, disease rates, and industrial companies.

### 4.3. Ontology Rules and Decision Making

Thus, we developed our ontology by defining relationships between different modules to provide a spatiotemporal and legal context for assessing water quality in the study area. Figure 2 shows an overview of the developed ontology network maintained by Protégé [42] and expressed in OWL2. The ontology rules were built based on the SWRL and SSN.

Rule 1: The hasWaterHygieneStandards rule establishes the pollution classes of water objects based on the normative rules of the Water Code of the Republic of Kazakhstan. Classes are determined based on the WPI—this indicator is set by national regulatory authorities depending on the water object and its location. For water bodies of the Republic of Kazakhstan, the following parameters were set for determining the WPI, as shown in Table A3 (see Appendix A):

As noted above in Section 4.1, observations of the quality of surface waters in the Almaty region were carried out at 42 gates of 22 water bodies, and when studying surface waters, 44 physical and chemical quality indicators were determined in the selected water samples. Monitoring of the quality of bottom sediments and coastal soil was carried out at 14 control points of the Ili River and Lakes Balkhash and Alakol [5].

Based on these indicators, the WPI was calculated, based on which Rule 2 was determined.

Rule 2: The *hasWaterClass* rule is used to determine the suitability/unsuitability of water for domestic and drinking water use (Table A1). Sensory observations record the results of samples in the form of chemical substances and their concentrations. With the help of Rule 2, the *Water_Class* class is automatically determined, where the WPI limits are specified in the ontology. Figure 8 shows the setting of WPI limits for the *Water_Class*.

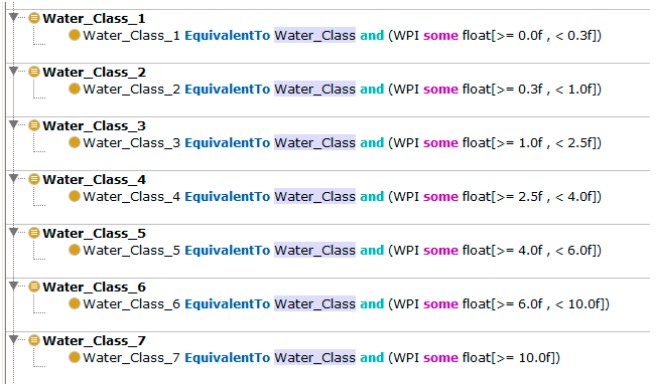

**Figure 8.** Populating the Water_Class with WPI Limits.

Rule 3: The *OD:Surface Water Resource* rule describes surface water resources with parameters such as the code of hydrological posts, water level, water temperature, water discharge, ice thickness and snow depth on ice, high water, and rain flood information. Based on this rule, statistical indicators are determined, for example, based on observation data, it is possible to calculate the maximum/minimum values of water discharges,

information about recurring floods at a certain time, and at which hydrostations critical or abnormal values are observed.

Rule 4: The *hasWaterObjects* rule associates sensor observations with water objects from the Observed Objects Module. Here, water bodies are divided into 4 types: lakes, reservoirs, rivers, and canals. Each water body, depending on the type, has parameters such as the area of a lake or reservoir, the length of rivers or canals, a hydrological resource, and an energy resource. With this rule, the data from the sensors are divided into groups, and the user can compare the flow and water level readings with the total volume or length of the water body.

Rule 5: The *hasTime* rule establishes temporal information that can be used to analyze temporal relationships, perform temporal queries, and support various temporal analyses and applications: *tl:Instant* represents a specific moment in time; *tl:Interval* represents a time interval or duration; and *tl:TemporalEntity* represents a generic temporary entity. This rule associates an observation with the time it was made or an event with a specific interval of time. The user can also define properties for temporal comparisons.

Rule 6: The *hasLocation* rule performs coordinate binding of water objects to regions. The regions of Kazakhstan have socioeconomic indicators: population, birth/death rate of the population, disease rates, and industrial companies. This is one of the most important rules, because it can be used to identify the correlation between the pollution of water bodies and the incidence of health issues in the population, as well as to determine other similar correlations.

### 4.4. Spatial Information Module

The spatial information module is designed for excellent and fast analysis by decision makers using the field of view. This module uses cutting-edge technologies and ontology-based knowledge to visually display important information, offering an intuitive and comprehensive overview of water resources in the Ili-Balkhash basin, considering various contextual factors.

To achieve this goal, the coordination of hydropower plants in this basin is organized strategically using ontological structures. This collaborative effort has made it possible to seamlessly integrate water level and discharge data at each hydrostation onto a geographic map.

In addition to hydrological data, our spatial information module includes the socioeconomic indicators needed to better understand the impacts of water management. In particular, we are integrating population statistics and water pollution index (WPI) scores into our spatial visualization. This inclusion allows users to explore not only the hydrological variables, but also the socioeconomic aspects surrounding each hydrostation. The result is an interactive map that visually illustrates the changing state of water resources across the basin, as shown in Figure 9. To represent WPI on the map, we use a color-coding system to ensure clarity and quick understanding. Circular markers are used: light green variants represent WPI levels of classes 1–2, dark green for classes 2–3, yellow for classes 3–4, and red for classes 4–5. This visual categorization allows stakeholders to quickly assess the severity of water pollution in various regions of the Ili-Balkhash basin.

Moreover, given that the Ili-Balkhash basin spans four regions of Kazakhstan, our spatial representation dynamically adapts to regional differences. Each region's color scheme adjusts based on population density, providing a detailed understanding of how socioeconomic factors interact with water resources in different geographic regions.

This innovative tool provides decision makers with an integrated view of water resources, promoting sound sustainable management strategies while highlighting the complex interactions between water and society in the Ili-Balkhash basin.

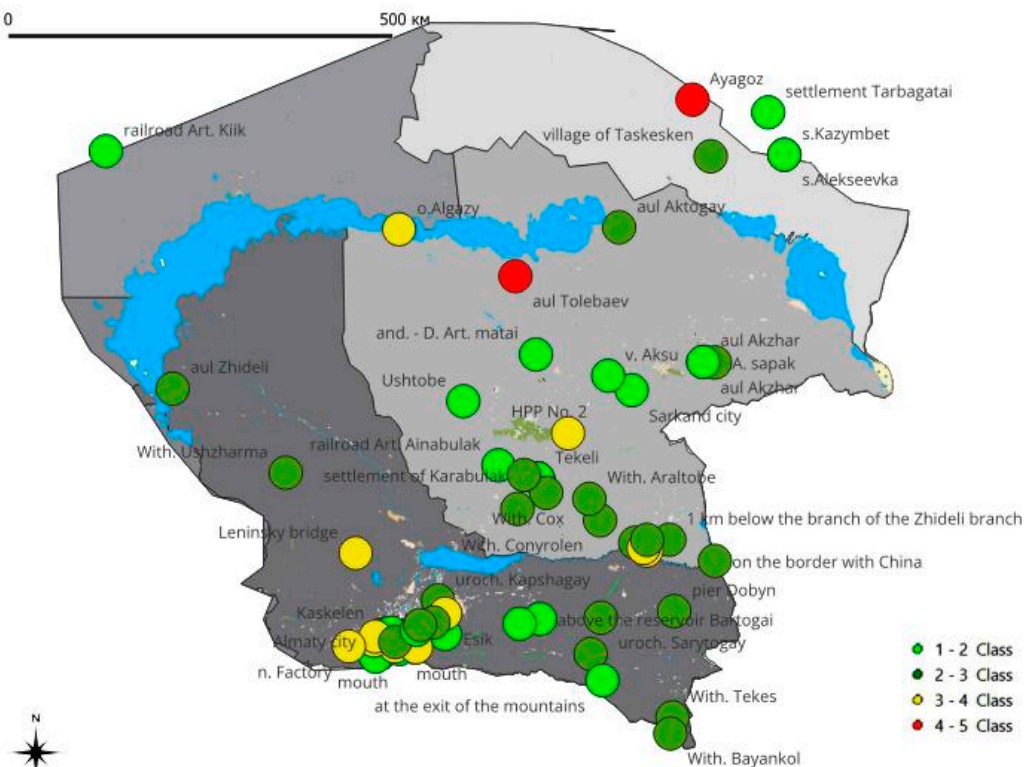

**Figure 9.** Spatial visualization of data on the population of the regions and classes of water pollution of the IBB hydro–stations.

## 5. Results

The schematic structure of the ontology was implemented in Protégé, as shown in Figure 10. So, at first, we created a skeleton; that is, we defined the main classes and their subclasses: KZ Regions, Sensor Observation, Timestamp, Water Class, Water Hygiene standards, and Water Objects.

Further, using the owlready library [43], the ontology was imported into Python. This library provides the ability to create, modify, load, and save ontologies, as well as perform various operations with them, such as searching for class instances and checking instance properties. Then, our data are imported, having already been reformatted into XML format. Next, the ontology is analyzed, which consists of the following steps:

1. Analysis of objects;
2. Property analysis;
3. Relationship analysis;
4. Analysis of classes of objects.

During these stages, the lists and dictionaries listed below are completed:

```
object_names = []
data_properties = []
object_properties = []
class_names = []
name2object = {}
name2data_property = {}
name2object_property = {}
```

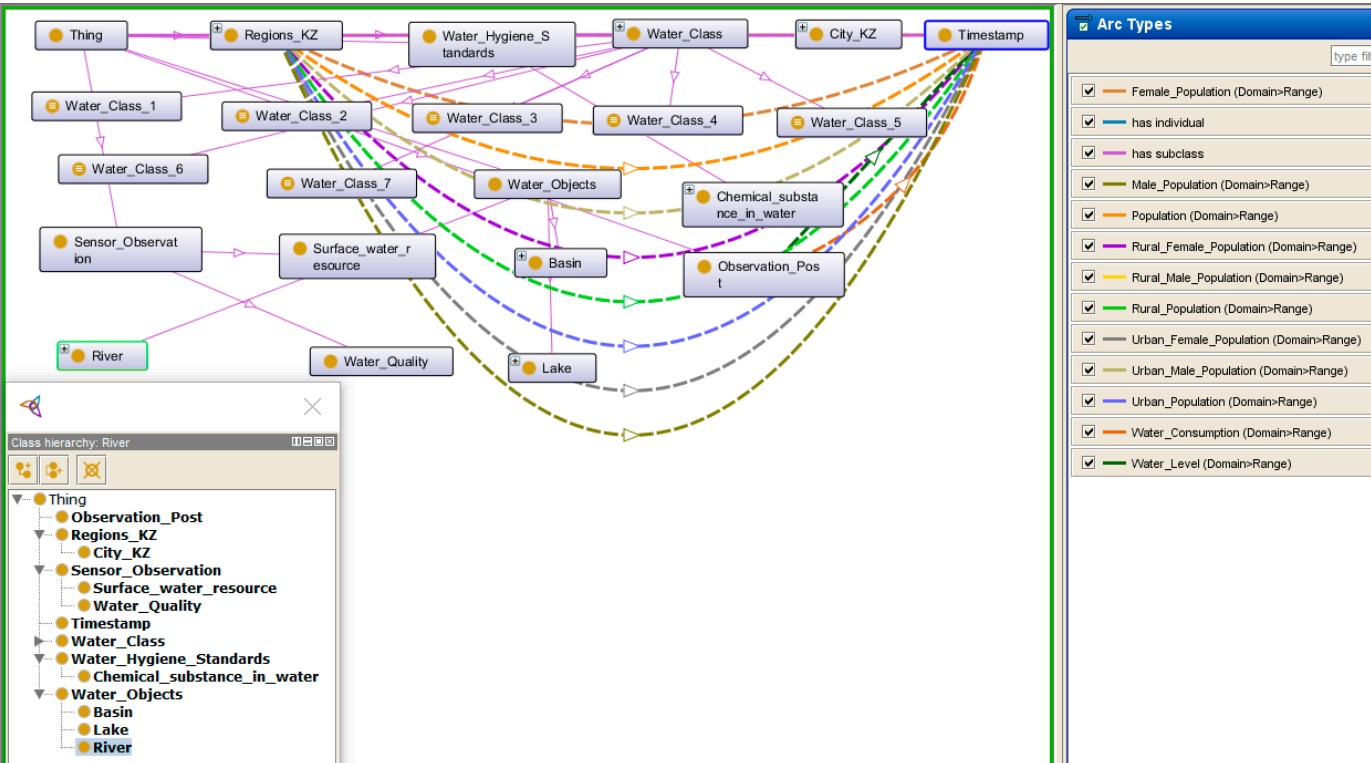

**Figure 10.** Schematic structure of the ontology implemented by using Protégé.

The first stage is the analysis of ontology objects, during which all objects from the ontology are extracted, and their names are normalized (reduced to lower case) and entered into the "object_names" list. Also, a pair is entered into the "name2object" dictionary: the name of the object is a link to the object. These auxiliary data structures are needed to facilitate the task of parsing tables. Subsequent modules use the "object_names" list to find objects and then use "name2object" to obtain a reference to the object in the ontology, for example, to add a new property to the object. Using a link to an object allows the user to change the object directly in the ontology. Similar actions are carried out for properties (data properties) and relations (object properties). For properties and relationships, accessing the object by reference is necessary to validate the property. As a result, we obtain a populated ontology with object properties, data properties, and individuals and logical relationships between them, and the results are shown in Figure 11.

After populating the ontology, it is necessary to check the correctness of the related data. To do this, we choose our research area—IBB. When checking for the Balkhash basin and the Ili river, only 7 object properties are defined for Balkhash and Ili, which show the following relationships: Balkhash is located in the Almaty and Zhambyl regions, representing the implementation of the links between the Regions and Water Objects classes; in the Balkhash basin, there are the Charyn, Kurty, Ile, and Chelek, reflecting an implementation of Basin River subclass links; in addition, 12 properties are defined that point to numeric attributes, as can be seen in Figure 12.

The experiment involved exporting OWL axioms to the rule engine through the SWRL Tab tool in Protege. Subsequently, Drools was used to execute the rules and infer new axioms based on the provided ontology. The entire process was timed to analyze its efficiency. The results of the knowledge transfer process and inference are shown in Table A4 (see Appendix A):

The data indicate that the knowledge transfer process from OWL to the rule engine was successful, with a substantial number of OWL declarations and axioms being seamlessly exported. This suggests that the SWRL Tab tool efficiently facilitated the translation of OWL knowledge to SWRL rules for use in the Drools rule engine.

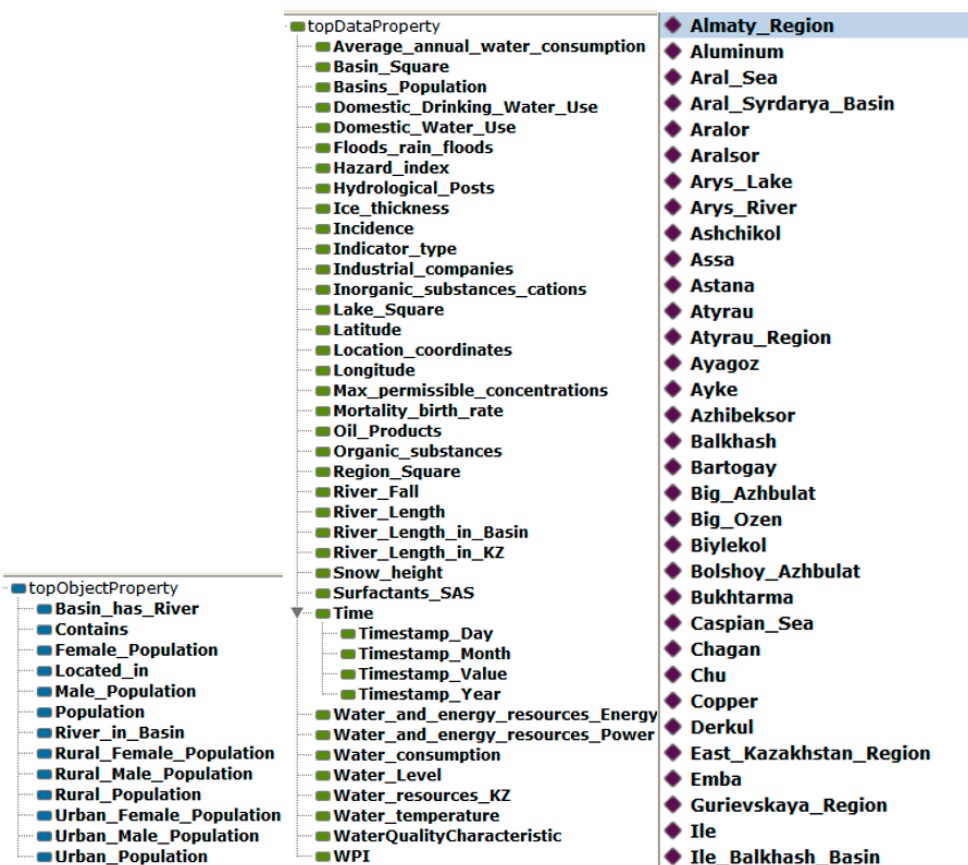

**Figure 11.** Object properties, data properties, and individuals automatically populated using owlready.

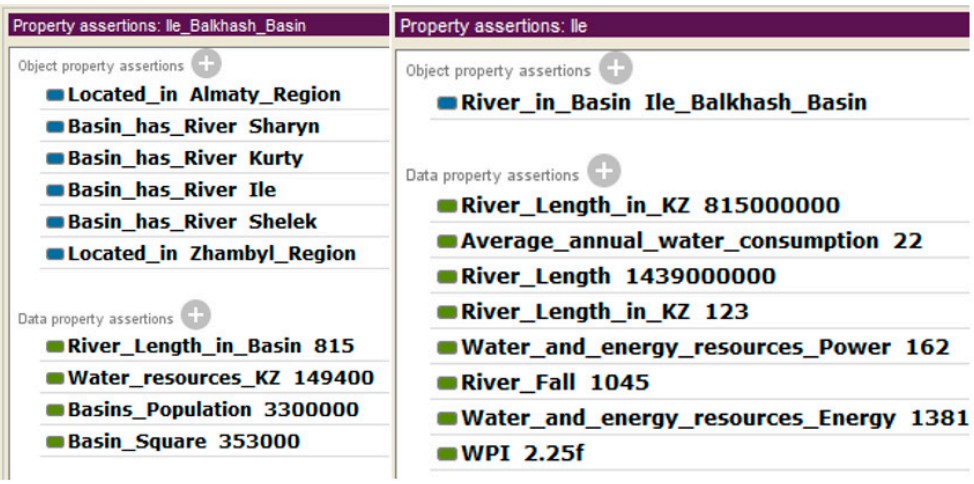

**Figure 12.** Defined object properties and data properties for IBB.

Because the main goal of creating this ontology is to establish the impact of the quality and quantity of IBB water resources on the socioeconomic situation of the region to which they belong, Figure 13 shows all the associated object properties for the Almaty region, because the IBB is mainly located here. For the Almaty region, first of all, social indicators were determined, such as the urban population, the rural population, the number of women/men, the life expectancy, and the morbidity rates for this region, including diseases of the circulatory system associated with iodine deficiency, malignant neoplasms, and acute infections of the upper respiratory tract. The impact of water quality on the health of the population living near large water objects was proven in [22]. Based on this, we can draw a

conclusion from our data about the relationship between the incidence of health issues in the population and the quality of the water resources.

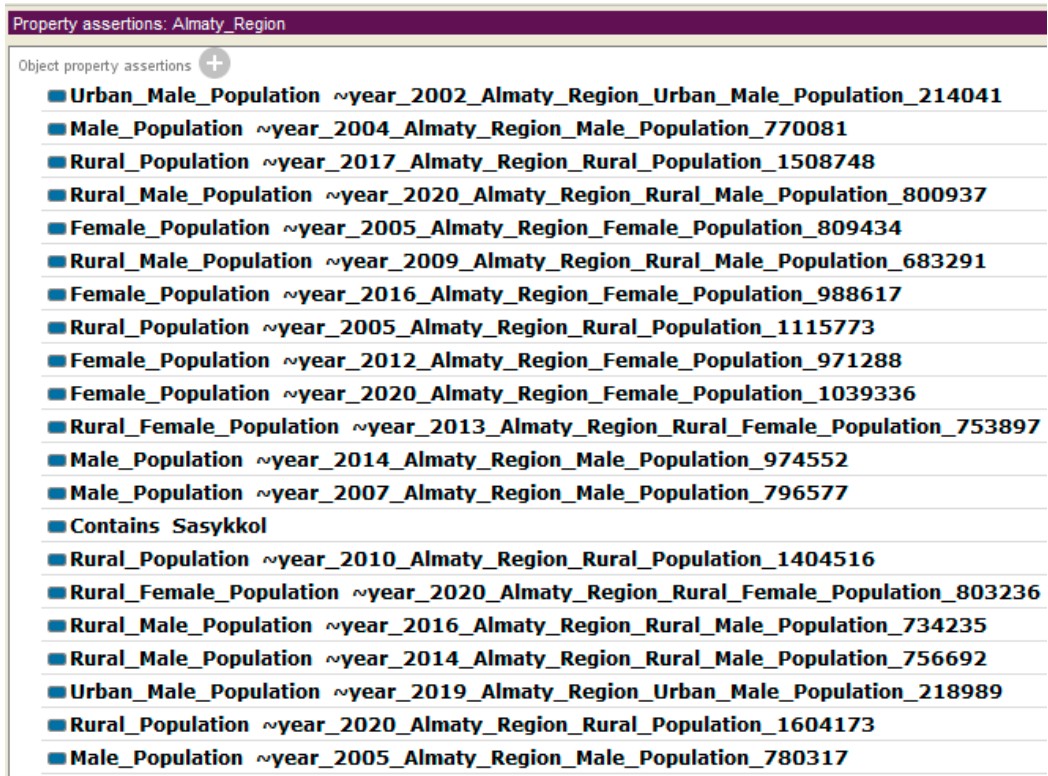

**Figure 13.** Related object properties for the Almaty region.

Now that the ontology has been populated and logical relationships have been established, we needed to implement queries in SPARQL that will help us derive new knowledge. For our ontology, the following queries were implemented:

1. Query 1 implements the derivation of water objects, the regions to which these objects belong, and the population of this region in the time interval, as shown in Figure 14.
2. Query 2 displays water objects and their WPI indicators, as well as their pollution class based on WPI indicators. The query result is shown in Figure 15.
3. Query 3 displays water objects with given WPI indicators provided that 0 < WPI < 5, as shown in Figure 16.
4. Query 4 displays water bodies with high WPI values and the number of people suffering from diseases of the circulatory system associated with iodine deficiency. The query result is shown in Figure 17.

Thus, based on the created ontology, it is possible to implement many queries to obtain useful knowledge that we could not attain if the data were stored in tables or other files.

Because we are using a temporal ontology, we can query and retrieve information based on temporal criteria, such as retrieving data within a specific time range or finding events that happened before or after a given timestamp.

Figures 18 and 19 show graphs of the results for queries 1 and 4. WPI indicators are indicated on the left along the coordinate, while the population and the number of people suffering from circulatory diseases are on the right. Here it can be noted how the water pollution index is related to the population and the incidence of health issues in the population. In future work, a thorough analysis based on machine learning will be carried out to identify direct correlations between water pollution and the incidence of the population.

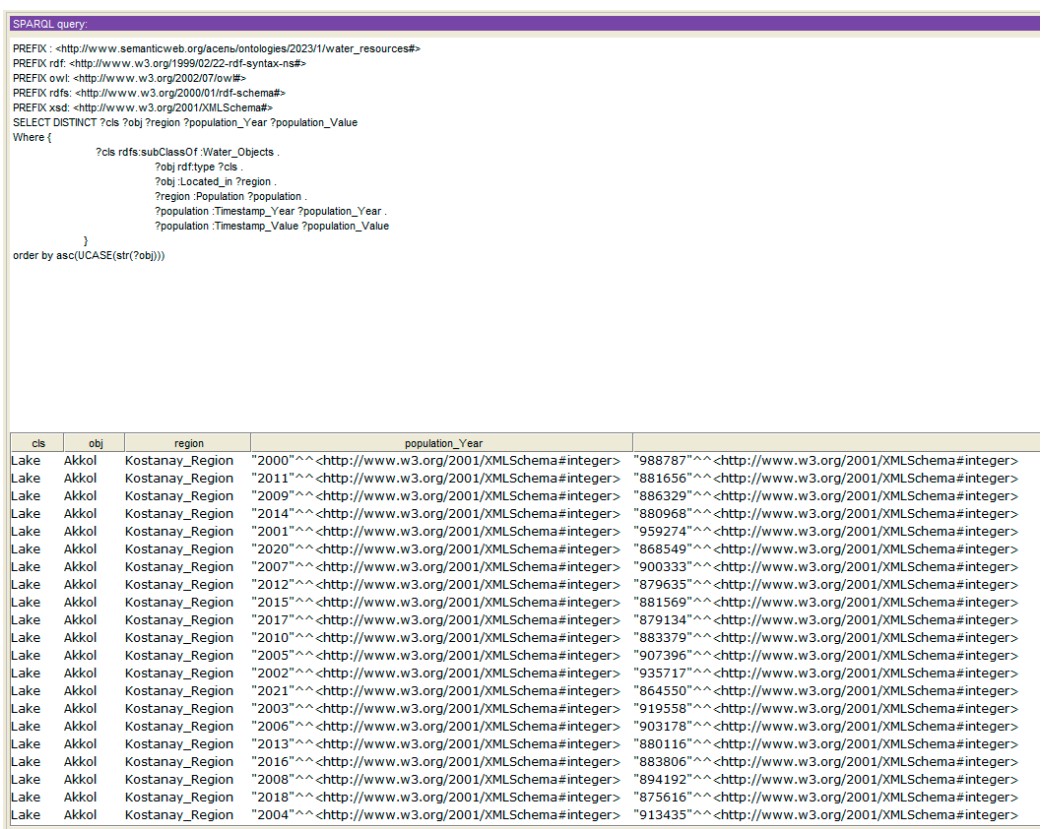

**Figure 14.** Query for displaying water objects, regions to which these objects belong, and population of this region in the time interval.

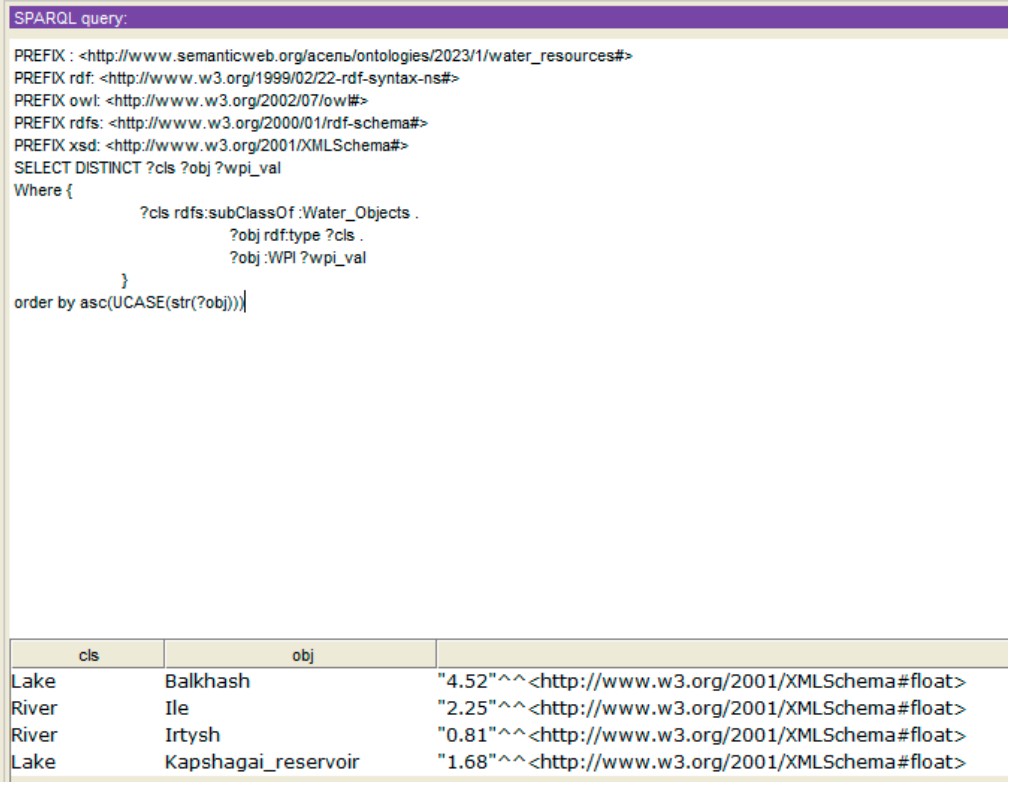

**Figure 15.** Query for displaying water bodies and their WPI indicators.

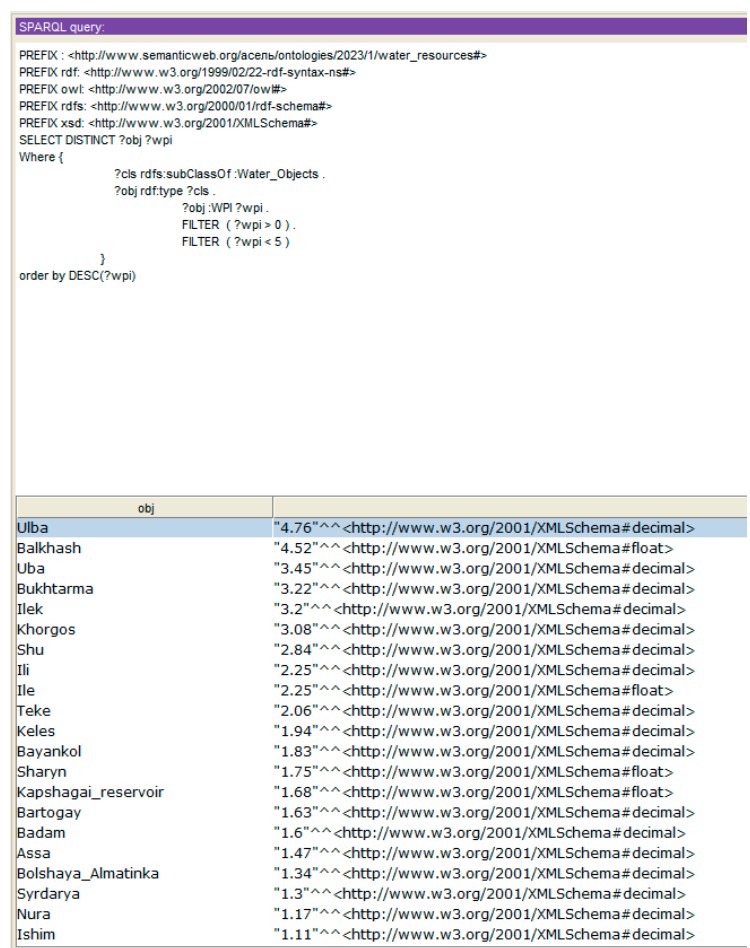

**Figure 16.** Query for inferring water bodies with given WPI indicators.

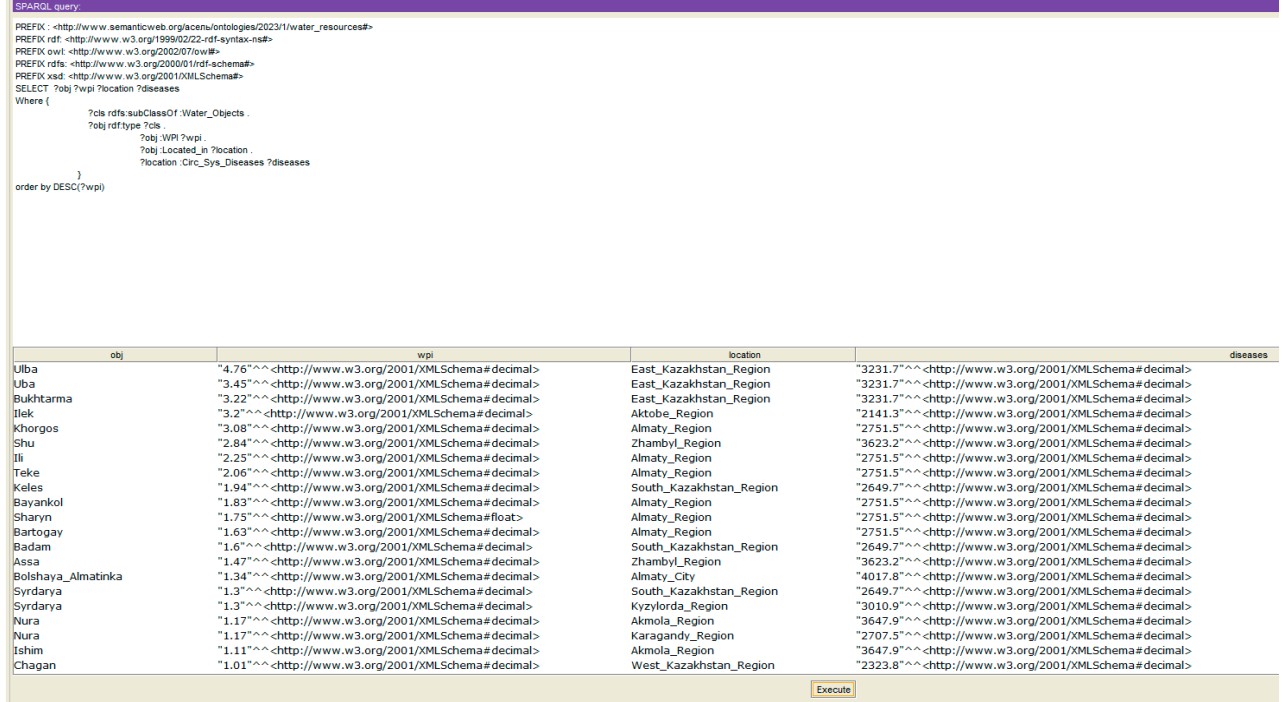

**Figure 17.** Result of the query for the deriving of water objects with high WPI values and the quantity of people suffering from diseases of the circulatory system associated with iodine deficiency.

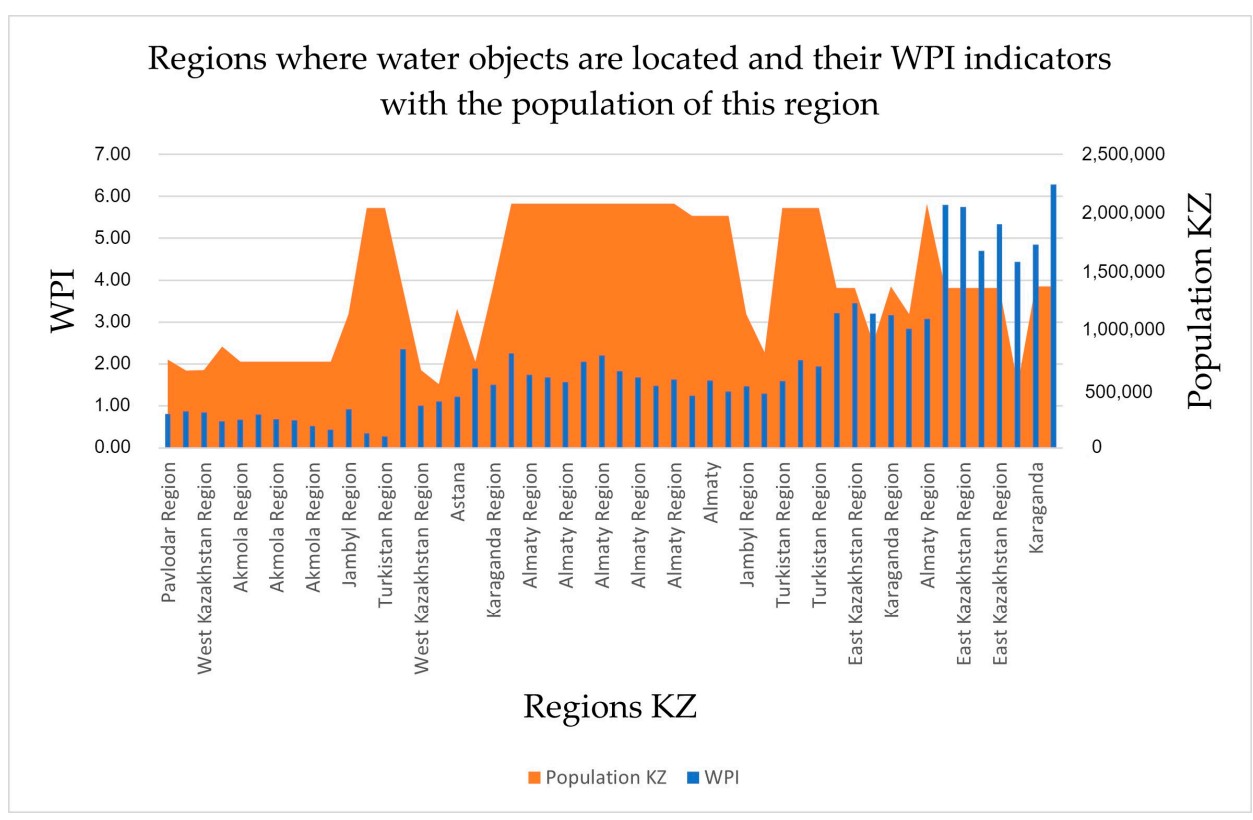

**Figure 18.** Visualization of query 1.

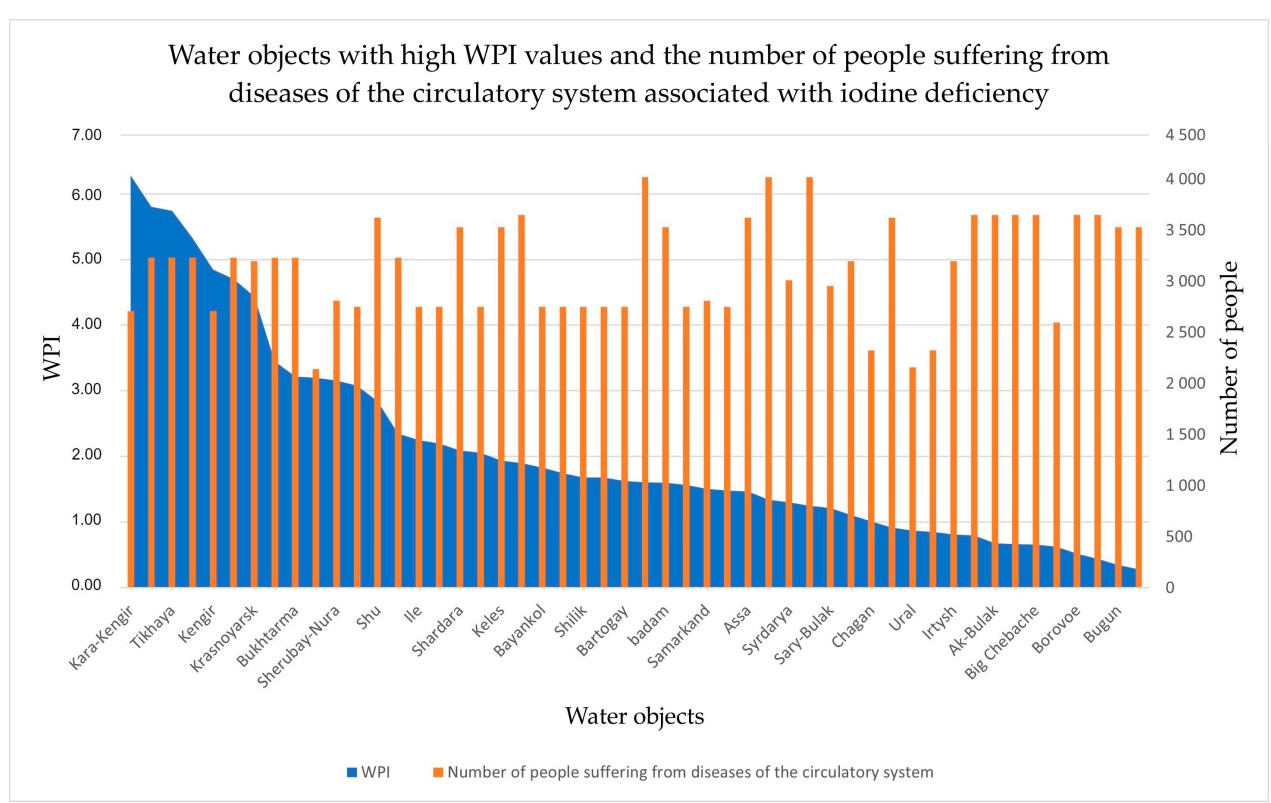

**Figure 19.** Visualization of query 4.

The research findings resulting from the implementation of SPARQL queries on our meticulously developed ontology for water resource monitoring in Kazakhstan hold

paramount importance in enabling effective water management and informed decision making. By leveraging these queries, we are able to glean profound insights and thoroughly evaluate the current status of water resources, pollution levels, and their potential ramifications on the populace. The salience of these queries lies in their capacity to furnish comprehensive and indispensable information for the sustainable management of water resources.

Query 1 has yielded fruitful results, facilitating the derivation of water objects in conjunction with their corresponding geographical regions, as well as the populace inhabiting these regions during specific time intervals. This pivotal information stands to greatly enhance our comprehension of water resource distribution across diverse geographic areas, thereby facilitating an evaluation of the potential ramifications of water scarcity or pollution on local communities. Such profound insights afford policymakers the opportunity to concentrate their efforts on specific regions that may necessitate additional water management strategies or resource allocation to ensure ecological equilibrium and human welfare.

Through query 2, we gained an intricate portrayal of water objects, featuring their respective Water Pollution Index (WPI) indicators and the concomitant pollution classes they belong to. This comprehensive assessment of water quality empowers us to discern areas where pollution levels may be elevated, compelling the implementation of remedial measures. Moreover, the standardized WPI indicators enable seamless comparisons and longitudinal monitoring, engendering a data-driven approach to safeguarding water quality.

Query 3 yielded significant outcomes, revealing water objects with WPI indicator values ranging between 0 and 5, indicative of regions with moderate to low pollution levels. Such findings are of utmost consequence in identifying locales with superior water quality and can potentially serve as benchmarks to inspire and propel other regions toward enhancing their water management practices and pollution control strategies.

Query 4 has emerged as a pivotal revelation, bringing to light water bodies with heightened WPI values, indicative of severe pollution. Additionally, it provides crucial data on the incidence of circulatory system diseases linked to iodine deficiency among the affected population. This information serves as a critical assessment of the health implications of water pollution, underscoring the urgency for prompt intervention and remediation to safeguard the well-being of our citizens.

In conclusion, the comprehensive outcomes derived from these SPARQL queries offer compelling evidence of the tremendous significance of our ontology in the realm of water resource monitoring. By virtue of this ontology and the associated queries, decision makers are equipped with a profound understanding of water quality, its spatial distribution, and its potential repercussions on public health. Informed by this knowledge, policymakers and water resource managers can chart targeted strategies, implement effective pollution control measures, and ensure the sustainable management of water resources for the holistic well-being of our environment and our populace in Kazakhstan. Moreover, the adaptability and scalability of our ontology render it a valuable resource for addressing analogous water resource challenges in other regions, thereby contributing to global water security and the pursuit of sustainable practices.

## 6. Discussion

The key question addressed in this study is the development of an ontology for water resources as a tool for studying sustainable regional development. By developing an ontology specific to environmental tasks, particularly the assessment of sustainable water use, the aim was to enhance understanding of the complex interrelationships among environmental factors, human activities, and the Sustainable Development Goals (SDGs) adopted by the United Nations General Assembly in 2015 [44]. For instance, Goal 6, "Clean Water and Sanitation", from this list entails:

Ensuring access to clean water and sanitation services for all people.

Ensuring the sustainable use and management of water resources, including the preservation of related ecosystems, the protection and restoration of aquifers, reservoirs, and water ecosystems.

Reducing water pollution and improving water quality, including the reduction in harmful chemical discharges, and improving wastewater treatment.

Improving water resource efficiency, reducing water losses in various sectors, including agriculture, industry, and urban infrastructure.

Protecting and restoring ecosystems associated with water resources, such as rivers, lakes, aquifers, and wetlands to maintain their ecological integrity and diversity.

In this section, we discuss the significance of the obtained results for monitoring the achievement of the mentioned goals. First and foremost, the ontology adapted to the context of water resources proved to be a valuable tool for organizing and structuring knowledge. The ontology serves as a conceptual framework that captures the interconnections and interdependencies among various entities, such as water bodies, economic entities, sources of pollution, humans, flora, and fauna. With the help of the ontology, we were able to create a knowledge base that facilitated the integration of diverse datasets and enabled efficient extraction of the required information. We developed sample queries to the knowledge base that allowed us to assess the impact of human activities on the qualitative and quantitative indicators of water resources.

One of the significant advantages of ontology-based approaches is the ability to perform advanced data analysis and logical reasoning. By applying semantic analysis methods, we were able to uncover hidden relationships and derive new knowledge from existing data. For example, using the ontology, we discovered previously unnoticed connections between classes of water pollution and the increase in the population's incidence of circulatory system diseases, as well as between the loss of biodiversity in river fauna and classes of river pollution. This finding highlights the potential of ontological analysis in uncovering complex ecological patterns and making data-driven decisions.

Moreover, the ontology-based approach facilitates knowledge sharing and collaboration among researchers, policymakers, and stakeholders. By providing a shared vocabulary and a common understanding of ecological concepts, the ontology serves as a bridge across different disciplines and enables effective communication. This aspect of collaboration is particularly relevant in the context of regional sustainable development planning, where interdisciplinary efforts and stakeholder engagement are crucial. The ontology acts as a central knowledge repository, promoting interdisciplinary collaboration and supporting a holistic approach to sustainable development goals.

In [44], a knowledge organization system based on an ontology is proposed, which models the key elements of the United Nations' global system of Sustainable Development Goal indicators. This system currently includes 17 goals, 169 targets, and 231 unique indicators, along with over 450 related sets of statistical data supported by the global statistical community for monitoring progress toward the SDGs and the dataset containing these elements. In addition to formalizing and establishing unique identifiers for the components of the SDGs and their indicator system, the ontology includes mapping each goal, target, indicator, and dataset to their corresponding terms and subjects in the United Nations Bibliographic Information System (UNBIS) and the EuroVoc Thesaurus, facilitating multilingual semantic search and content linking.

As noted by the authors of this work, in order to promote a holistic approach through coordinated policies and actions involving governments of different countries and stakeholders from all levels of society, it is crucial to develop tools that facilitate the discovery and analysis of interrelationships among various global SDG indicators derived from different data sources, information, and knowledge supported by different stakeholder groups. Essentially, this is an attempt to provide stakeholders with a means to publish data using shared terminology and URIs centered around SDG concepts, which helps to enhance the semantic compatibility of diverse data and information related to SDG information assets provided by various societal layers. In this work, we contribute to the formation of such an

information asset, providing indicators and data for SDG 6 "Clean Water and Sanitation" at the regional level. Although we have not yet aligned our ontology with the system proposed by the authors, this task represents a technical step that is significantly facilitated by employing an ontological approach.

However, it is important to acknowledge the limitations and challenges associated with ontology development in the domain of water resources. Developing a comprehensive and accurate ontology requires significant domain knowledge and substantial effort. The process of ontology development involves iterative refinement and validation, which can be time-consuming and resource-intensive. Additionally, supporting and updating the ontology is necessary to ensure its relevance and alignment with new ecological knowledge. Therefore, future research should focus on automatically expanding the ontology to cover additional aspects and further validating its usefulness in real-world applications.

## 7. Conclusions

In summary, this study contributes significantly to the burgeoning field of ontology-based sustainable water resource management. The uniqueness of our research stems from the fact that, to the best of our knowledge, we have pioneered the development of a comprehensive pipeline that spans from the initial collection of diverse primary data sources (including sensors, national statistical repositories, and gauge data) to population and ontology alignment, query generation, and the realization of a decision support system complete with statistical dashboards and spatial visualization capabilities. Our system is constructed using RDF and boasts seamless integration potential into broader ontology frameworks.

Thus, we have successfully developed a knowledge graph based on ontologies using SWRL, SSN, and time ontology methods for monitoring the water resources of the Ili-Balkhash basin. By applying these methods, we have obtained an effective tool that allows us to formalize new knowledge and rules, integrate data from various sensors and devices, provide more comprehensive and accurate information about the state of the water resources, and account for temporal aspects, which is important for analyzing changes over time.

A particularly noteworthy element is the inclusion of a spatial information module, specifically designed to facilitate expedient and highly effective decision making. Utilizing cutting-edge technologies, the module synergistically combines with our ontology framework to offer a visually intuitive and comprehensive overview of the Ili-Balkhash basin's water resources. Furthermore, an important step was the data collection from heterogeneous sources using tools like Qurma, TableProcessor, and TableMiner. These tools enabled us to extract information from different data formats and transform it into a structured form, facilitating the integration of data into the knowledge graph. The result of this research is the acquisition of new knowledge about the level, discharge, and quality of water resources, which significantly impact the socioeconomic indicators of each region where these water bodies are located.

Our work holds significant potential for assessing the real-world impact in the field of water resource management. Beyond generating actionable insights about water levels, flows, and quality, the spatial information module offers decision makers an incredibly effective tool for rapid, in-depth analyses. As we look to the future, our aim is to integrate machine learning techniques to offer even more robust capabilities for predictive analysis, such as identifying future epidemiological zones, anticipating floods and droughts, or suggesting actionable measures for sustainable water resource management.

While acknowledging the challenges and limitations, such as the need for iterative ontology refinement and specialized domain knowledge, our future work will focus on the automated expansion of both the ontology and the spatial information modules.

In conclusion, our study represents a groundbreaking contribution to the field of water resource management. With its robust ontology and cutting-edge spatial information

modules, our work offers an advanced, dynamic, and multifaceted tool designed to meet the complex challenges of sustainability in this critically important domain.

**Author Contributions:** A.O. developed an ontology model and rules, collected data, participated in the development of parsers for extracting useful knowledge, performed an experiment, and wrote this paper. M.M. suggested ideas and research methods, participated in the development of Qurma and TableProcessor tools for data extraction, and advised in writing this article. V.B. suggested ideas and methods for creating an ontology, participated in the creation of the TableProcessor tool for extracting data from Excel files, suggested the idea of using the integration of socioeconomic indicators with data on water resources, and advised on writing this article. A.N. is the main author of the creation of the Qurma parser for extracting useful knowledge from PDF files, suggested ideas and research methods, wrote a discussion in this paper, and advised on writing this article. R.T. enriched the XML ontology with data and performed the tasks of implementing queries in SPARQL. All authors have read and agreed to the published version of the manuscript.

**Funding:** This study was funded by the Ministry of Science and Higher Education of the Republic of Kazakhstan, grant number AP09261344 "Development of methods for automatic extraction of spatial objects from heterogeneous sources for information support of geographic information systems".

**Institutional Review Board Statement:** Not applicable.

**Informed Consent Statement:** Not applicable.

**Data Availability Statement:** All tables utilized in this study are available in the "Tables" folder of our GitHub repository, which can be accessed via the following link: https://github.com/Titrom0 25/PyTableMiner/tree/main/Tables (accessed on 21 September 2023). Additionally, the updated ontology relevant to our research is stored in the "ontologies" directory of the same repository. Direct access to the ontology data is available at: https://github.com/Titrom025/PyTableMiner/tree/m ain/ontology (accessed on 21 September 2023). For those interested in a broader overview of the project's structure and components, the main web application template can be reviewed at this location: https://github.com/Titrom025/PyTableMiner/blob/main/webApp/templates/index.html (accessed on 21 September 2023).

**Conflicts of Interest:** The authors declare no conflict of interest.

## Appendix A

**Table A1.** Determination of Water Classes by WPI indicators.

| Class | Water Quality Characteristic | Water Pollution Index (WPI) | Domestic and Drinking Water Use | Domestic Water Use |
|---|---|---|---|---|
| I class | "very clean" | 0.0–0.3 | quite suitable | suitable |
| II class | "clean" | 0.3–1.0 | suitable | suitable |
| III class | "moderately polluted" | 1.0–2.5 | suitable for cleaning | suitable |
| IV class | "polluted" | 2.5–4.0 | not suitable | not suitable |
| V class | "dirty" | 4.0–6.0 | not suitable | not suitable |
| VI class | "very dirty" | 6.0–10 | not suitable | not suitable |
| VII class | "extremely dirty" | >10 | not suitable | not suitable |

**Table A2.** Heterogeneous data sources.

| Heterogeneous Data Sources | Source | Content |
|---|---|---|
| I Water Regulations | Order on Approval of the Sanitary Rules "Sanitary and Epidemiological Requirements for Water Sources, Places of Water Intake for Domestic and Drinking Purposes, Domestic and Drinking Water Supply and Places of Cultural and Domestic Water Use and Safety of Water Bodies" [8] | 1. General Provisions;<br>2. Sanitary and epidemiological requirements for water sources;<br>4. Indicators of drinking water quality;<br>5. Microbiological and parasitological indicators of drinking water quality;<br>6. Hygienic standards for the content of harmful substances in drinking water;<br>7. Quantity, frequency of water sampling;<br>8. List of indicators. |

**Table A2.** *Cont.*

| Heterogeneous Data Sources | Source | Content |
|---|---|---|
| II Sensor Data | Daily hydrological bulletin of the Republic of Kazakhstan [5] | 1. Location of hydrological posts;<br>2. Water level;<br>3. The state of the water object;<br>4. Water temperature;<br>5. Weather conditions;<br>6. Water consumption;<br>7. Thickness of ice and height of snow on ice;<br>8. Ice phenomena at the site of the post;<br>9. Information about floods and rain floods. |
| | Monthly State of the Environment Newsletter [6] | 1. The main sources of air pollution;<br>2. The state of the quality of atmospheric air;<br>3. The chemical composition of atmospheric precipitation.<br>4. The state of the quality of surface waters;<br>5. Radiation environment. |
| III Water Objects | Information about IBB from Wikipedia [9] | 1. Physicogeographical description;<br>2. Soils and vegetation;<br>3. Hydrography;<br>4. Glaciers;<br>5. Hydropower resources;<br>6. Knowledge of river flow;<br>7. Economic activity. |
| IV Socioeconomic Indicators | Bureau of National Statistics of the Republic of Kazakhstan [7] | 1. Population;<br>2. Birth/mortality of the population;<br>3. Diseases of the circulatory system associated with iodine deficiency;<br>4. Malignant neoplasms;<br>5. Acute infections of the upper respiratory tract;<br>6. Life expectancy;<br>7. Types and activities of industrial companies. |

**Table A3.** Hygienic standards for the contents of chemicals in water to determine WPI.

| № | Substance_Name | Standards (MPC), Not More than in mg/L |
|---|---|---|
| 1 | Total mineralization (dry residue) | >1000 |
| 2 | General hardness | >7.0 (mg-eq./L) |
| 3 | Oil products, total | >0.1 |
| 4 | Surfactants (SASs), anionic | >0.5 |
| 5 | Inorganic substances (cations) | Depends on the type of chemical substance |
| 6 | Organic substances | Depends on the type of chemical substance |

**Table A4.** The results of the knowledge transfer process and inference.

| № | Name of Process | Values |
|---|---|---|
| 1 | Number of OWL class declarations exported to the rule engine | 21 |
| 2 | Number of OWL individual declarations exported to the rule engine | 180 |
| 3 | Number of OWL object property declarations exported to the rule engine | 16 |
| 4 | Number of OWL data property declarations exported to the rule engine | 41 |
| 5 | Total number of OWL axioms exported to the rule engine | 725 |
| 6 | Number of inferred axioms | 843 |

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
