# Peer review of "The Development of a Water Resource Monitoring Ontology as a Research Tool for Sustainable Regional Development"

_data, 2023_

Round 1

Reviewer 1 Report

Comments and Suggestions for Authors

See the attachment

Comments on the Quality of English Language

Moderate editing of English language required

Author Response

Dear Reviewer,

We would like to express our sincere gratitude for taking the time to review our manuscript and for your constructive comments and suggestions. Your feedback has been invaluable in improving the quality of our work. We have taken into account all your comments and suggestions. Please check attached file.

Reviewer 2 Report

Comments and Suggestions for Authors

The Development Of The Monitoring Water Resources Ontology As A Research Tool For Sustainable Regional Development

The paper is a good one, and probably very helpful for the authors' country, but I am very surprised by the fact that key terms used hard by practitioners in various trades are missing (e.g. dispatcher chart, used for decades in water management, etc. .).

Political interference with science is unacceptable! Never have the most eminent scholars reached high political office, so please refrain!

In figure 1, the term "hydrological post" must be corrected with that of "hydrometric station or gauging station". A hydrological location means more than observation, measurement and monitoring ... Also, symbols related to lake hydrometric stations can be made in blue, that nothing bad happens ... Lakes have special regimes within water resources and require differentiation...

The work lacks the most important element, related to a national level dispatch system: the graphic transposition of the information (interactive thematic maps), for an excellent and quick analysis of the decision-making factors, with a view to intervention. So, an information spatialization module is mandatory, which presents the information in real time. For the authorities the interface must be total, and for the general public partial. This ontology will, at some point, be a national platform of great utility ...

The Conclusion chapter is very thinly constructed. Development is needed.

Author Response

Dear Reviewer,

We would like to express our sincere gratitude for taking the time to review our manuscript and for your constructive comments and suggestions. Your feedback has been invaluable in improving the quality of our work.  We have taken into account all your comments and suggestions. Please check attached file

Round 2

Reviewer 1 Report

Comments and Suggestions for Authors

The authors have revised the manuscript according to the suggestions

Comments on the Quality of English Language

Minor editing of English language required

Author Response

Dear Reviewer,

Thank you for taking the time to review our manuscript and providing your valuable feedback. We sincerely appreciate your suggestion regarding the need for minor English editing.
In response to your recommendation, we have sought professional assistance to ensure the language and grammar quality of our manuscript meets the standards expected. Specifically, we have utilized the English language checking services provided by the MDPI journal to ensure clarity, coherence, and correctness in the revised manuscript.
We believe that with this professional touch, our manuscript now offers a clearer and more polished presentation of our research. We remain open to any further suggestions you might have, and once again, we thank you for your keen observation and guidance.

Warm regards,

Assel Ospan